# A numerical study on melt water feedback in the coupled Arctic Sea ice-ocean system

Haohao Zhang[1, 2], Xuezhi Bai[1, 2], Kaiwen Wang[1, 2]

[1]Key Laboratory of Marine Hazards Forecasting, Ministry of Natural Resources, Hohai University, Nanjing, 210098, China

[2]College of Oceanography, Hohai University, Nanjing, 210098, China

Correspondence to: Xuezhi Bai (xuezhi.bai@hhu.edu.cn)

**Abstract.** A one-dimensional coupled sea ice-ocean model is used to investigate the effects of the melt water feedback on sea ice. In the control experiments, the model is capable of accurately simulating seasonal changes in the upper ocean stratification structure compared to observations, and the results suggest that ocean stratification is important for ice thickness development during the growing season. The sensitivity experiments reveal the following: 1) A decrease in melt water release weakens ocean stratification and creates a deeper, higher salinity mixed layer. 2) Melt water release has negative feedback on ice melting, reducing ice melting by 16.6% by strengthening ocean stratification. 3) The impact of melt water released during the previous melting season on ice growth in winter depends on the strength of stratification. After removing all the melt water during the summer, ice formation in areas with strong stratification increased by 12.3% during the winter, while it decreased by more than 40% in areas with weak stratification. 4) In some areas of the Nansen Basin where stratification is nearly absent, the warm Atlantic water can directly reach the ice in early spring, leading to early melting of the sea ice in winter if all melt water is removed from the model. These findings contribute to our understanding of the complex interactions between ocean stratification, melt water, and sea ice growth and have important implications for climate models and future change prediction in the Arctic.

## 1 Introduction

The upper Arctic Ocean is strongly stratified with primarily ice coverage and a high volume of freshwater input (Rawlins et al., 2010; McClelland et al., 2012; Rudels, 2015). The Arctic Ocean consists of three main layers. The top layer is a cold and fresh surface layer. The intermediate layer is a cold halocline layer (CHL), which is characterised by gradually increasing salinity, and the bottom layer is a relatively warm and salty Atlantic Water (AW) layer. This stratification pattern is crucial for the existence of Arctic Sea ice, as the fresh surface layer and CHL protect the ice cover from the heat stored in the AW layer below (Rudels et al., 1996a; Steele and Boyd, 1998; Martinson and Steele, 2001; Rudels et al., 2005). Freshwater flux from river runoff, positive net precipitation, relatively fresh Pacific inflow, and seasonal ice melt are critical factors that maintain this stratification (Haine et al., 2015; Carmack et al., 2016).

Ocean-ice heat fluxes play a crucial role in modulating the Arctic Sea ice growth/melt cycle, with half of the total heat flux absorbed by the sea ice originating from the ocean (Carmack et al., 2015). According to a one-dimensional coupled ice-ocean model study by Toole et al. (2010), the very strong density stratification at the base of the mixed layer (ML) in the Canada Basin greatly impedes surface layer deepening and thus limits the flux of deep ocean heat to the surface, which could influence sea ice growth and decay. Linders and Björk (2013) note that ocean stratification is mostly important for ice growth during the growing season because areas with weak stratification have larger ocean-ice heat fluxes, resulting in less ice formation during winter. Davis et al. (2016) use a one-dimensional model to show that the sea ice in the Eurasian Basin is more sensitive to changes in vertical mixing than that in the Canada Basin due to its weaker ocean stratification.

Ice melting is particularly important for seasonal changes in stratification and ocean-ice heat fluxes in the Arctic Ocean (Jackson et al., 2010; Toole et al., 2010; Linders and Björk, 2013; Hordoir et al., 2022). The volume of net input freshwater that comes to the Arctic Ocean from external sources is roughly $1200 \pm 730$ km$^3$ annually, with an inflow of $9400 \pm 490$ km$^3$yr$^{-1}$ and an outflow of $8250 \pm 550$ km$^3$yr$^{-1}$, while approximately 11300 km$^3$ freshwater enter the ocean in summer through melting. The volume of melt water in the Arctic Ocean is approximately 10 times greater than that of net freshwater input, leading to an increase of 1.2 m in the Arctic Ocean's surface freshwater layer (Haine et al., 2015), which separate the surface ML from the near-surface temperature maximum (NSTM). In winter, surface fresh water is recycled via ice formation, weakening ocean stratification (Peralta-Ferriz and Woodgate, 2015), meanwhile, vertical convection caused by brine rejection or storm-driven mixing, can erode the NSTM layer, entraining warm water upward, and impeding winter ice formation (Steele et al. 2011; Jackson et al. 2012; Timmermans, 2015; Smith et al., 2018).

Melt water from the sea ice has a comparatively low density and therefore accumulates in the top ocean layer, strengthens the upper ocean stratification. Due to the stabilizing of the cold halocline, the ocean heat flux available to melt sea ice decreases, which in turn hinders sea ice melting. This is a negative sea ice/ocean feedback on sea ice melting (Bintanja et al., 2013), and we call it melt water feedback in this paper. Zhang (2007) and Bintanja et al. (2013) suggest that this negative sea ice/ocean feedback can explain the anomalous increase in Antarctic sea ice extent before 2010s. Many positive feedback processes in the Arctic atmosphere-ocean-ice systems are extensively studied, such as the well-known sea ice albedo feedback (Hall, 2004; Winton, 2000; Pithan and Mauritsen, 2014), water vapor feedback (Gordon et al., 2013; Taylor et al., 2013) and the Cloud-Albedo feedbacks (Zelinka et al., 2012; Bodas-Salcedo et al., 2016). However, there are almost no quantitative studies on this negative melt water feedback on sea ice melting in the Arctic, although many previous studies have investigated the effects of increased freshwater flux by adding freshwater flux to the ocean surface in models to represent increased runoff or precipitation (Nummelin et al., 2015, 2016; Davis et al., 2016a; Pemberton and Nilsson, 2016).

To enhance the comprehension of the feedback mechanisms of melt water on sea ice, we use a one-dimensional coupled sea ice-ocean model and modify the source code to control the release of melt water to the ocean to quantitatively assess the responses of ocean and sea ice to different amounts of melt water release to the ocean. One-dimensional models have been

widely used in previous studies of the Arctic Ocean's vertical structure and ice cover (Killworth and Smith, n.d.; Price et al., 1986; Bitz et al., 1996; Björk, 2002a, b; Peterson et al., 2002; Linders and Björk, 2013; Nummelin et al., 2015, 2016; Davis et al., 2016a). A one-dimensional model is simplistic because it does not take into account advection processes; however, it usually provides a reasonable simulation of upper ocean stratification that matches observations well in short simulation time (Toole et al., 2010; Linders and Björk, 2013).

Additionally, the intensity of stratification varies across the Arctic Ocean, with a gradual weakening from the Canada Basin towards the Eurasia Basin. In the Canada Basin, a lower-saline upper layer results in a well-developed and persistent cold halocline (Toole et al., 2010). In contrast, the cold halocline layer is quite weak or even absent in some areas of the Eurasin Basin (Rudels et al., 1996b; Steele and Boyd, 1998; Björk, 2002b), such as those close to Svalbard in the Nansen Basin, where the warm AW are more easily mixed upwards and reach the ice cover (Rudels et al., 2005). Previous research suggests that brine-driven surface convection could entrain the AW heat upwards in the Eurasian Basin (Polyakov et al., 2013a, 2020), while the strong stratification impede this convection process in the Canada Basin (Toole et al., 2010). Given the considerable spatial variability in the stratification strength across the Arctic Ocean, the impact of melt water is expected to vary regionally. Thus, this study investigates regional variations in the effect of melt water on the ocean and sea ice by experimenting with the initial temperature and salinity profiles from multiple stations in the Arctic Ocean.

The paper is organised as follows: Section 2 details the model setup and the sensitivity experiments. Section 3 presents the model results and discusses how the ocean and sea ice respond to reduced melt water release. A discussion is provided in sections 4. Section 5 reviews the conclusions.

## 2 Model description and sensitivity experiments

### 2.1 Coupled Sea ice-ocean model

We use a one-dimensional, coupled sea ice-ocean model based on the Massachusetts Institute of Technology general circulation model (MITgcm) (Marshall et al., 1997) to explore the melt water feedback in a coupled ice-ocean system in the Arctic Ocean. The water column in the model extends from the surface down to a depth of 300 m, and the vertical grid has a uniform thickness of 1 m. The ocean model utilises the nonlinear equation of state of Jackett and McDougall (1995) and the nonlocal K-Profile Parameterization (KPP) vertical mixing scheme of Large et al. (1994). Shaw and Stanton (2014) show that the vertical diffusivity in the deep central Canadian Basin averages near-molecular levels, ranging between $2.2 \times 10^{-7}$ m$^2$ s$^{-1}$ and $3.4 \times 10^{-7}$ m$^2$ s$^{-1}$, and Fer (2009) found that vertical diffusivity between $10^{-6}$ -$10^{-5}$ m$^2$ s$^{-1}$ in the Eurasian Basin. The background vertical diffusivity of the model used in this study is set to $10^{-6}$ m$^2$ s$^{-1}$, which is a representative value in the central Arctic Ocean and has been applied to several one-dimensional models studying the Arctic Ocean (Linders and Björk, 2013; Nummelin et al., 2015; Davis et al., 2016)

The sea ice package is based on a variant of the viscous-plastic sea ice model (Losch et al., 2010) and is combined with the thermodynamic sea ice model of Winton (2000) and Bitz and Lipscomb (1999). Although the one-dimensional model includes a dynamics sea ice module, sea ice changes are only determined by thermodynamics processes. The model considers two equally thick ice layers: the upper layer has a variable specific heat resulting from brine pockets, and the lower layer has a fixed heat capacity. The heat fluxes at the ice top and bottom are:

$$F_{top} = F_s(\alpha) - Fs_{ice} \tag{1}$$

$$F_{bot} = Fb_{ice} - F_b \tag{2}$$

where $F_s$ is the surface heat flux absorbed by the ice, $Fs_{ice}$ is the conductive heat flux from the upper layer of the sea ice to the ice surface, $Fb_{ice}$ is the conductive heat flux from the ice bottom to the lower layer of the sea ice. $F_b$ is the ocean-ice heat flux:

$$F_b = c_{sw}\rho_{sw}\gamma(T_{sst} - T_f)u^* \tag{3}$$

where $\gamma$ is the heat transfer coefficient and $u^*$ is the frictional velocity between ice and water.

The albedo parameterization of this model is dependent on ice thickness (Hansen et al., 1983):

$$\alpha = \alpha_{i_{min}} + (\alpha_{i_{max}} - \alpha_{i_{min}})(1 - e^{-h_i/h_\alpha}) \tag{4}$$

where $\alpha_{i_{min}}$=0.1 and $\alpha_{i_{max}}$=0.64 are the maximum and minimum ice albedo values, respectively $h_\alpha$=0.65 is the ice thickness for albedo transition, and $h_i$ is the ice thickness.

The net ocean surface heat flux can be written simply as Steele et al., (2010):

$$F_{ocean} = F_{sw} + F_b + F_{ao} \tag{5}$$

where $F_{sw}$ is the heat flux from solar radiation, $F_b$ is the ocean to ice heat flux, and $F_{ao}$ is the heat flux from the ocean to the atmosphere through the ice-free area (including longwave radiation and sensible and latent heat flux).

## 2.2 Initial conditions

The model is initialised with a given ice thickness (2.5 m), ice concentration (95%), and time-averaged temperature and salinity profiles measured by Ice-Tethered Profiles (ITPs) (Krishfield et al., 2008; Toole et al., 2011). The data from 14 ITPs are selected as initial profiles in the model simulations: A1-A7 located in the Amerasian Basin (the blue dots in Fig. 1) and E1-E7 in the Eurasian Basin (the red dots in Fig. 1). Data from other 6 ITPs are used to evaluate the simulation (The black dots in Fig. 1), and they are all located close to the simulated stations. The details of the ITP records used in this study are listed in Table 1.

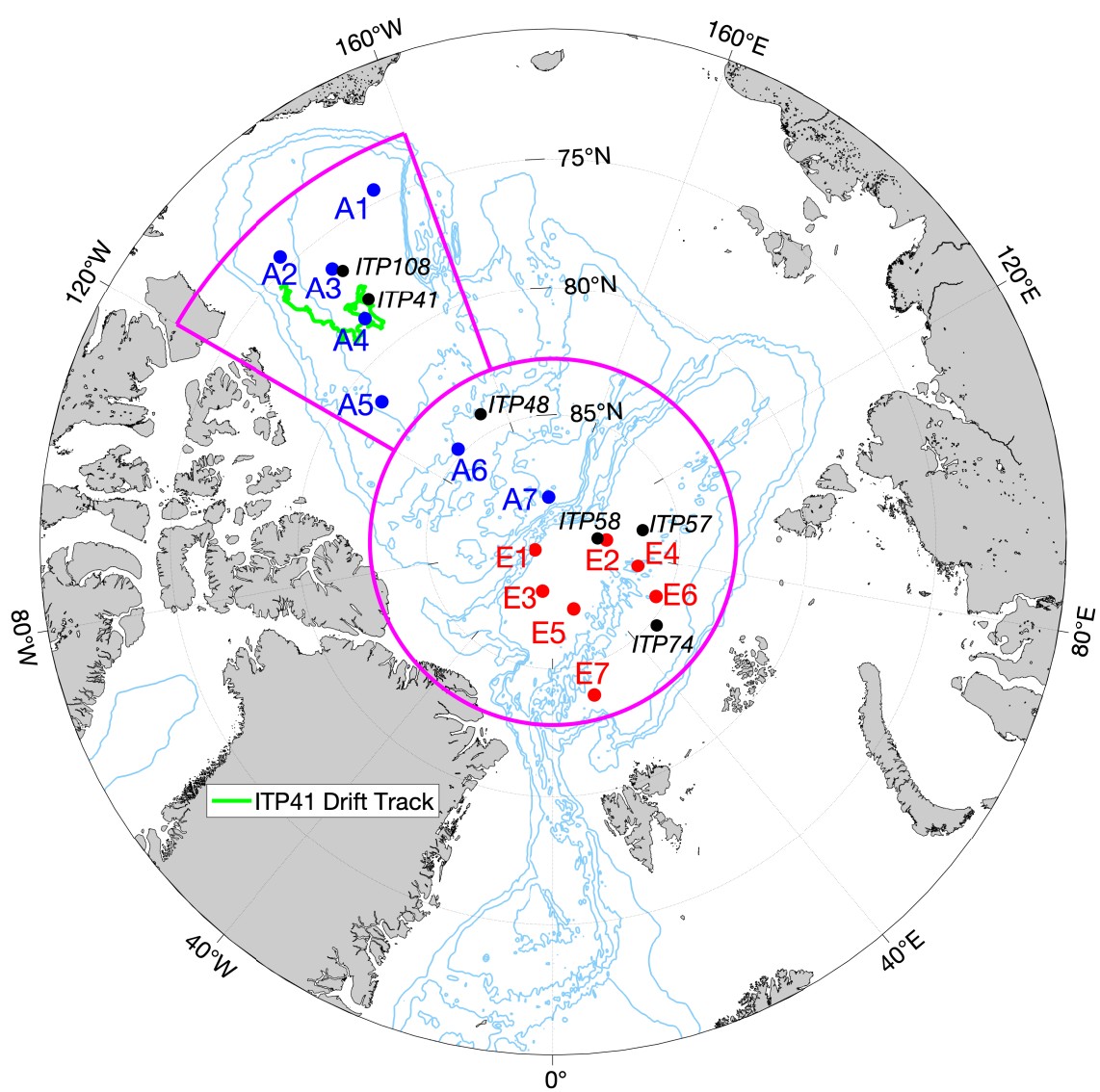

**Figure 1: Locations of the ITP data used as initial profiles in the model. Stations A1-A7 are located in the Amerasian Basin (indicated by the blue dots), and E1-E7 are in the Eurasian Basin (indicated by the red dots). The black dots represent the ITPs used for comparison with the simulations. The green line represents the trajectory of ITP41. The bathymetry is from ETOPO-2. The same atmospheric forcing field, derived from the 2011-2020 average for the specific region outlined by the solid magenta line, is utilized in all experiments.**

**Table 1. Details of the ITP records used in the model**

| Station | ITP number | Time | Comparison ITP number/Time |
|---------|-----------|------|---------------------------|
| A1 | ITP-53 | 2012.5.1-5.5 | |
| A2 | ITP-18 | 2008.4.6-4.7 | |
| A3 | ITP-108 | 2018.5.7-5.13 | ITP-108/2018.1.1-1.31 |
| A4 | ITP-41 | 2011.5.1-5.15 | ITP-41/2011.8.1-8.5 |
| A5 | ITP-105 | 2019.5.2-5.10 | |
| A6 | ITP-48 | 2012.5.9-5.23 | ITP-48/2012.1.18-1.31 |
| A7 | ITP-47 | 2011.4.12-4.30 | |
| E1 | ITP-93 | 2016.5.1-5.8 | |
| E2 | ITP-57 | 2013.5.25-5.28 | ITP-58/2013.3.9-3.10 |
| E3 | ITP-83 | 2015.5.25-5.30 | |
| E4 | ITP-74 | 2014.5.1-5.2 | ITP-57/2013.8.1-8.2 |
| E5 | ITP-58 | 2013.5.1-5.2 | |
| E6 | ITP-74 | 2014.5.26-5.30 | ITP-74/2014.8.1-8.5 |
| E7 | ITP-111 | 2020.5.25-5.30 | |

Figure 2 shows the time-averaged vertical profiles of the temperature, salinity, and buoyancy frequency from the 14 ITPs. The buoyancy frequencies show that the strength of ocean stratification gradually decreases from the Pacific side towards the Atlantic side (Fig. 2c and f). The vertical temperature profiles at A1, A3, and A4 stations show a temperature maximum at around 50m in the upper layer, which is the Pacific Summer Water (PSW) and is widely present in the central and western

Canadian Basin (Shimada et al., 2001; Steele, 2004). The temperature profile at station A2 shows two peaks in the upper layer, one is the NSTM, and the other is the PSW. The initial profile at A2 station was obtained from ITP measurements in the southern Canadian Basin in 2007-2008, and due to a strong halocline that year, the NSTM formed in the summer of 2007 persisted until the spring of 2008 (Jackson et al., 2012). Another noticeable feature is the temperature minimum observed around 175m in A1-A4 stations, which is the Pacific Winter Water (Fig. 2a). Stations A6 and A7 are in the Makarov Basin,

and the profiles show a transition feature from Pacific to Atlantic water influence. The upper layer of stations E1-E7 in the Eurasian Basin is characterised by a cold and fresh surface ML overlying a deeper warm (T > 0°C) and salty AW layer and weaker ocean stratification than the Amerasian Basin (Fig. 2d-f). Station E1, located at the Lomonosov Ridge, despite being closer to the Eurasian Basin, also has strong stratification features similar to those of stations in the Amerasian Basin. Stations E6 and E7, in the Nansen Basin, have much weaker salinity stratification than other stations in the Eurasian Basin. These

vertical profiles reflect various stratification features across the Arctic Ocean.

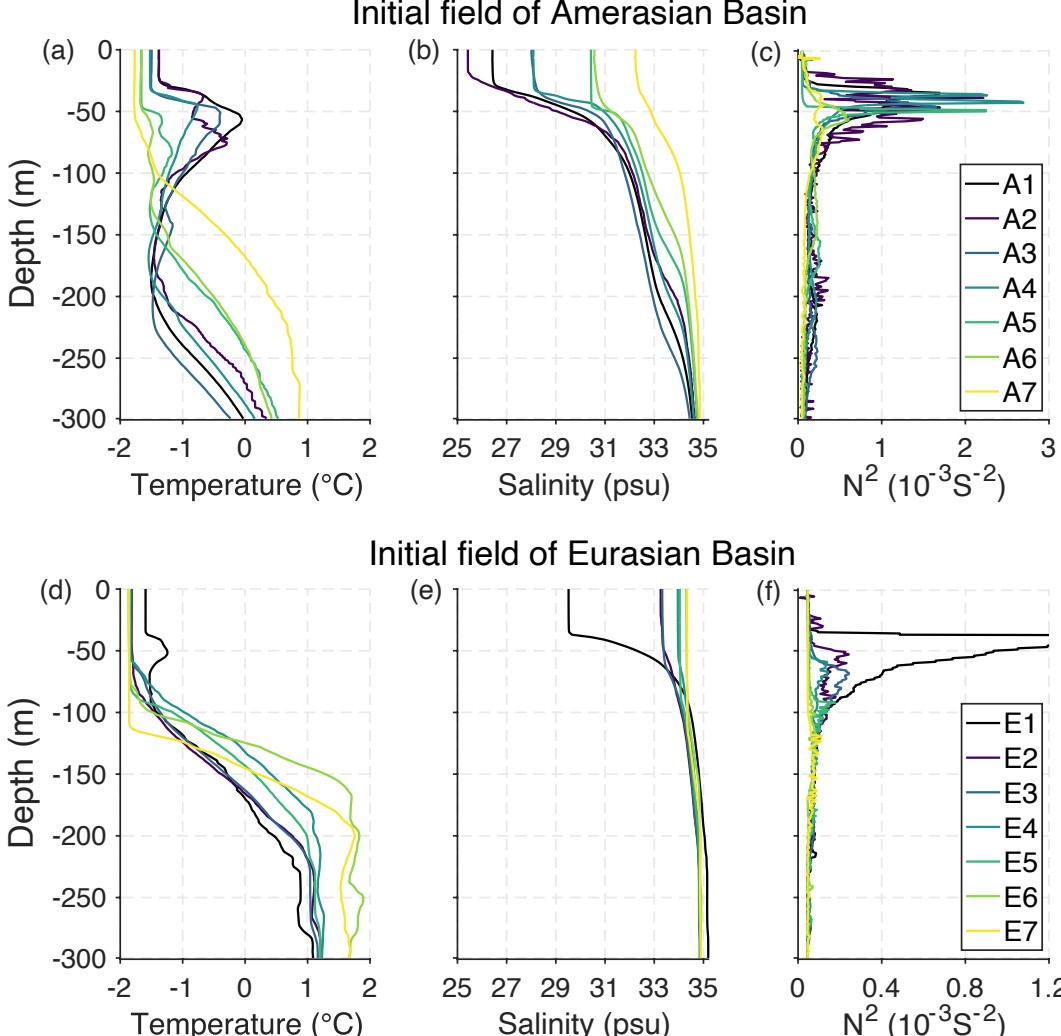

**Figure 2: The observed temperature (a, d) and salinity (b, e) profiles obtained from ITPs in the Arctic Ocean, which are used as the initial profiles in the model. The corresponding buoyancy frequency values (c, f) for each station are also displayed. The time of observation for each station is shown in Table 1.**

**2.3 Atmospheric forcing and freshwater input**

Atmospheric forcing for the model includes daily 10-m wind speed, 2-m air temperature, 2-m specific humidity, and downwards long- and shortwave radiation, from the National Centres for Environmental Prediction-Department of Energy (NCEP-DOE) Reanalysis 2, all of which are regionally averaged over the area delineated by the black boundary in Fig. 1. The averages are calculated over the period of 2011 to 2020 and cover the area defined by the two subareas spanned by 83–90°N

latitudes and 0–360°E longitudes (Central Arctic Ocean) and 73–83°N latitudes and 200–240°E longitudes (Canada Basin). The same atmospheric forcing is used for all model runs to eliminate the effects of differences in atmospheric forcing.

Although the focus of this study is on the melt water feedback in the coupled ocean-sea ice system, freshwater fluxes due to runoff inflow, precipitation minus evaporation, and input or output from straits also contribute to the stratification changes of the Arctic Ocean. The signal of the melt water feedback is to be exaggerated if those external freshwater forcings were ignored. So, we also consider the external freshwater forcing in the experiments. Haine et al. (2015) reported that the annual net inflow of freshwater to the Arctic Ocean is approximately 1200 km$^3$yr$^{-1}$, and we add this net freshwater inflow to our model on a daily average to represent various freshwater sources other than the melt water.

## 2.4 Sensitivity experiments

To investigate the impact of the release of melt water on ocean stratification and sea ice, a total of six experiments were conducted at each station for a simulation period of 1 year, starting on May 1 and ending on April 30 next year. The first is the control run, and the other five experiments are melt water perturbation (MWP) runs with 0%, 20%, 40%, 60%, and 80% melt water release into the ocean. The experiment started on 1 May with the objective of conducting a full melting period followed by a complete freezing phase in the model, which helps to better investigate the feedback effects of melt water on sea ice melting in summer, as well as its impact on subsequent freezing in winter. In the coupled ice-ocean model, the melt water flux of a timestep (600s) is determined by the freshwater content of the sea ice before and after a timestep. In its initial state, the freshwater content of the sea ice is as follows:

$$W_{frw} = \rho_{Ice} * H_{Ice} \qquad (6)$$

where $W_{frw}$ is the mass of fresh water initially present in the ice, $\rho_{Ice}$ is the density of the ice ($\rho_{Ice}$= 900 kg m$^{-3}$) and $H_{Ice}$ is the initial ice thickness. The melt water entering the ocean is calculated as follows:

$$F_{reflx} = (W_{frw} - \rho_{Ice} * h_{Ice})/\Delta t \qquad (7)$$

where $F_{reflx}$ (kg m$^{-2}$ s) is the ocean freshwater flux and $h_{Ice}$ is the ice thickness.

In the sensitivity experiments, we scale the freshwater flux by multiplying it by a factor $k$ to control the amount of melt water release:

$$F_{reflx} = k * \rho_{Ice} * (H_{Ice} - h_{Ice})/\Delta t \qquad (8)$$

We set $k$ to 0 (MWP-0% run), 0.2 (MWP-20% run), 0.4 (MWP-40% run), 0.6 (MWP-60% run), and 0.8 (MWP-80% run).

Figure 3a shows the time series of the ocean freshwater flux for the six experiments at station A1 as an example, in which the negative value represents freshwater entering the ocean. It is obvious that freshwater flux is negative (positive) during the ice-melting (ice-growth) season in the control runs. In the MWP runs, the freshwater flux is artificially reduced during the ice-melting season. As expected, the salinity gradient becomes weaker, and the ML deepens when the release of the melt water is reduced (Fig. 3b-g). In this paper, we define the base of the ML as the depth at which the potential density relative to 0 dbar initially surpasses the shallowest sampled density by the threshold criterion of $\Delta\sigma$=0.03 kg m$^{-3}$. Peralta-Ferriz and Woodgate (2015) used a threshold criterion of $\Delta\sigma$=0.1 kg m$^{-3}$ to define the ML and report the ML properties of the Arctic Ocean from

1979-2012. They also acknowledged that the Δσ=0.1 kg m$^{-3}$ criterion would overestimate the ML in parts of the Eurasian basin where the upper ocean's stratification is very weak. To account for this, they used a threshold of Δσ=0.03 kg m$^{-3}$ in areas with

weak upper ocean stratification. We calculated the simulated MLDs using the two methods, respectively, and found that in the Canadian Basin, the MLDs determined by the two criteria show no significant difference. However, in the Eurasian basin, the criterion of Δσ = 0.1 kg m$^{-3}$ would severely overestimate the MLDs. Therefore, we chose the criterion of Δσ = 0.03 kg m$^{-3}$ to determine the MLD in this study.

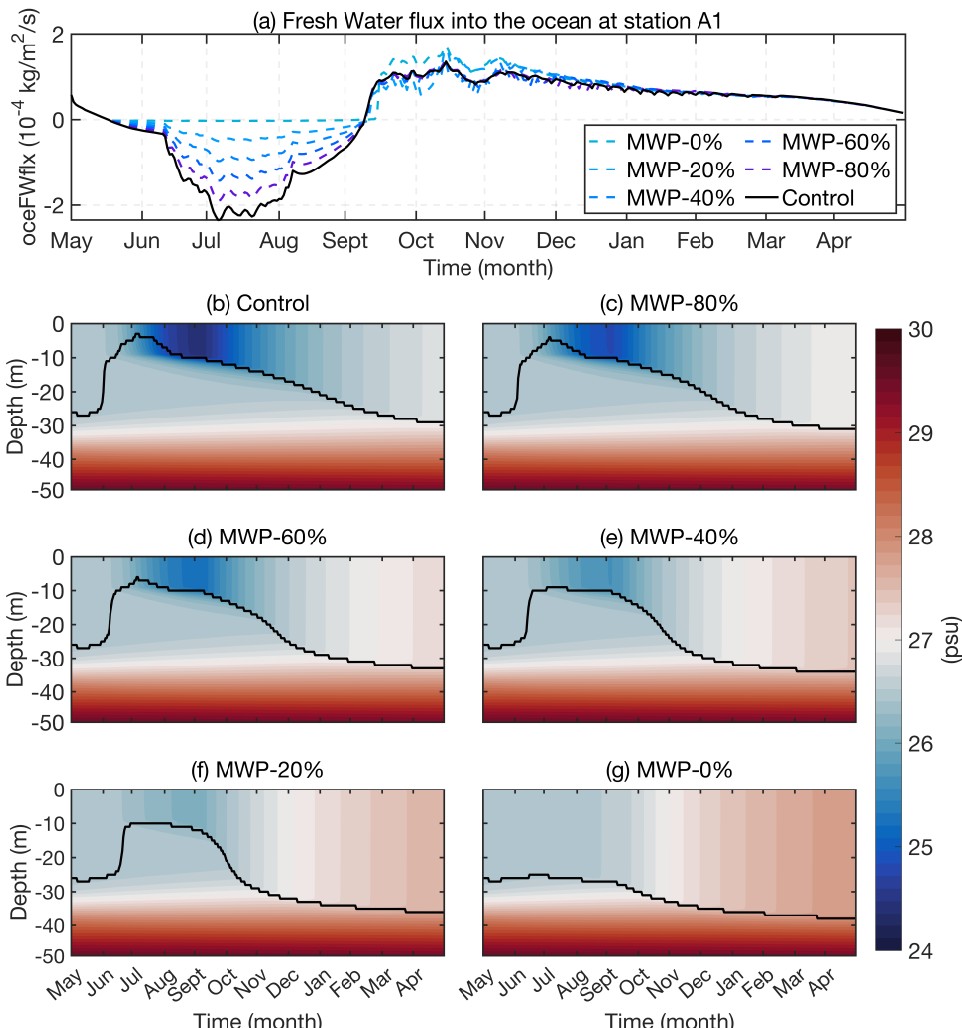

**Figure 3: Simulated sea surface freshwater flux and time series of upper 50 m salinity at station A1. (a): Time series of sea surface freshwater flux. The negative values represent the freshwater entering the ocean. In the legend, ~% refers to the magnitude of the melt water input anomaly in the MWP runs. (b): Time series of the upper 50 m salinity for the control run at station A1. (c)-(g): Time series of upper 50 m salinity for MWP runs at station A1. The black lines in (b)-(g) indicate the MLDs.**

## 3 Results

### 3.1 Control runs

#### 3.1.1 Upper Ocean thermohaline structure

Figures 4 and 5 show the comparison between the simulated temperature and salinity profiles of the control runs and the ITP observations (the details of the 6 ITPs datasets for comparison with the simulated results are listed in Table 1). The results of the one-dimensional model reasonably reproduce the seasonal variations of the vertical temperature and salinity structure in the Arctic Ocean. It should be noted that this study does not aim to perfectly replicate the variability of the ITP profiles, as the variability of the Arctic Ocean temperature and salinity structure is influenced not only by surface freshwater fluxes but also by an array of external local forcings, such as high-frequency variations in wind fields, local precipitation or evaporation, horizontal transport of freshwater, and observational errors. Despite some discrepancies between the simulated and observed vertical profiles, the simulations of these ideal experiments are still qualitatively consistent with the observations. Therefore, the simulation results obtained in this study are reliable.

ITP41 measured relatively complete temperature and salinity data along its pathway (green line in Fig. 1) in the Canadian Basin from May 2011 to April 2012, and the data measured by ITP41 in May 2011 also serve as the initial field for station A4 in the model. Therefore, we compared the complete time series of the temperature and salinity of the ITP41 observations with the simulations. Both the observations and simulations show that large quantities of freshwater, primarily melt water, cover the ocean surface during the melting season, typically lasting from June to September. As a result, a significant salinity gradient forms between the surface water and underlying water layers, creating a new, fresher surface layer (Fig. 4b and d). And the model also successfully reproduces the NSTM at the base of the summer ML, present at approximately 10-20 m (Fig. 4a and c). During the freezing season (October to the next April), brine rejection enhances the turbulence scale perturbations, leading to a deeper ML, and the NSTM generated during the summer progressively cools and vanishes (Fig. 4a and c).

Furthermore, we compared the simulated values with actual summer and winter observations gathered from select stations in the vicinity of the simulation. Figure 5 shows that the simulated vertical temperature and salinity profiles and MLDs for both summer and winter are similar to the nearby ITP observations. However, the simulated summer NSTM in the Canadian Basin is generally cooler than the observations (Fig. 5a). This discrepancy may lead to an overestimation of winter ice formation in the simulations. In all control runs, the simulated maximum winter MLD is ~33 m in the Canadian Basin, ~43 m in the Makarov Basin, ~67 m in the Amundsen Basin, and more than 100m in the Nansen Basin. These results are comparable to the observations. The observed maximum winter MLDs in Canada and the Makarov Basin are 29 ± 12 m and 52 ± 14 m, respectively, and those in the Eurasian Basin range from ~50 to over 100 m (Shimada et al., 2001; Peralta-Ferriz and Woodgate, 2015). Both the modelling and the observations show that the MLDs are usually deeper in the Eurasian Basin than in the Canadian Basin.

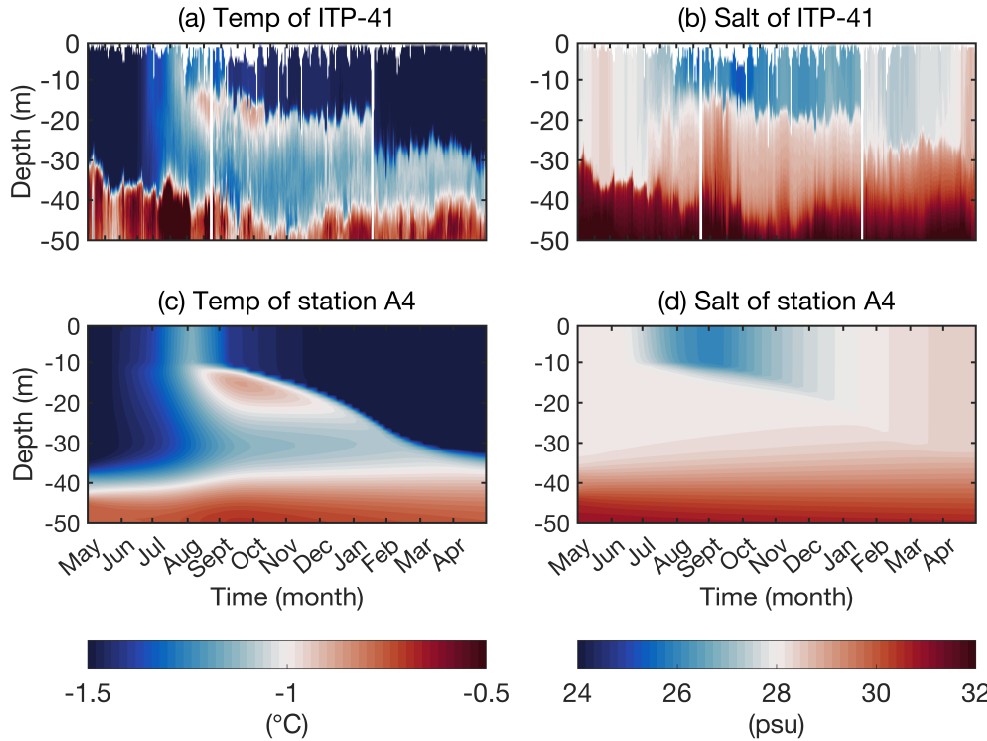


**Figure 4. The time series of temperature (left) and salinity (right) for the upper 50 m were derived from (a), (b): ITP-41 observations and (c), (d): simulated values at station A4, respectively. The trajectory of ITP-41 is shown in figure 1.**

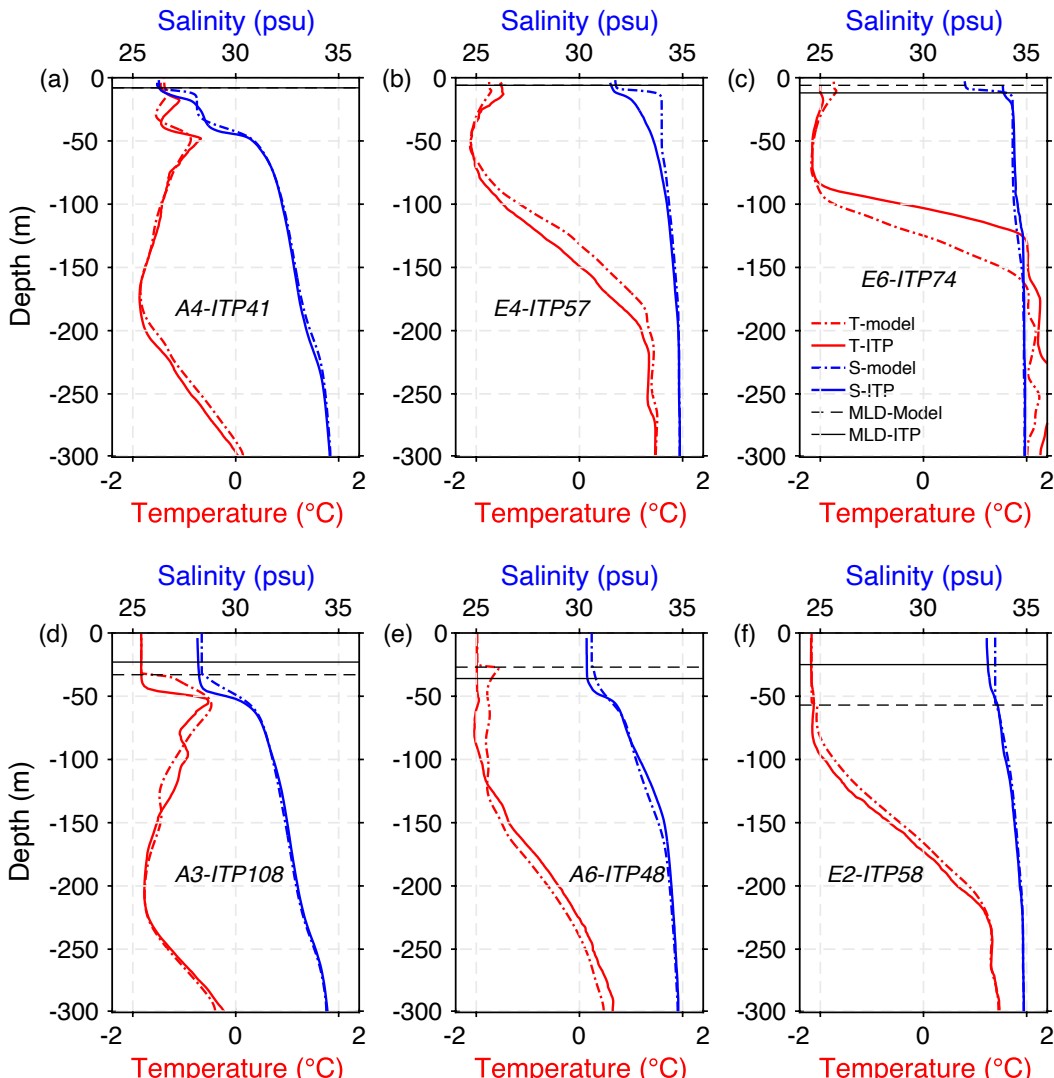

**Figure 5: Comparison of the simulated temperature (red line) and salinity (blue line) (dotted line) with the nearby ITP data (solid line) during summer (top row) and winter (bottom row). The depth of the ML is indicated by the black lines parallel to the x-axis. (a): A4 and ITP-41 in August. (b): E4 and ITP-57 in August. (c): E6 and ITP-74 in August. (d): A3 and ITP-108 in April. (e): A6 and ITP-48 in January. (f): E2 and ITP-58 in April.**

### 3.1.2 Sea ice and ocean-ice heat flux

Figure 6 shows the temporal development of ice thickness, ice concentration and ocean-ice heat flux in the control runs. The amount of ice melt during the melting season is basically independent of the initial ocean stratification. However, the sea ice growth from February to April shows dependence on the initial ocean stratification (subplot in Fig. 6a). Under the same atmospheric forcing, stations in the American Basin (A1-A7) with well-developed and persistent haloclines have more ice

growth (~1.68 m), while stations E6 and E7 in the Nansen Basin have less ice growth (~1.55 m) because the cold halocline is not fully developed there, and consequently, the higher ocean-ice heat flux from February to April inhibits ice formation (Fig. 6c).


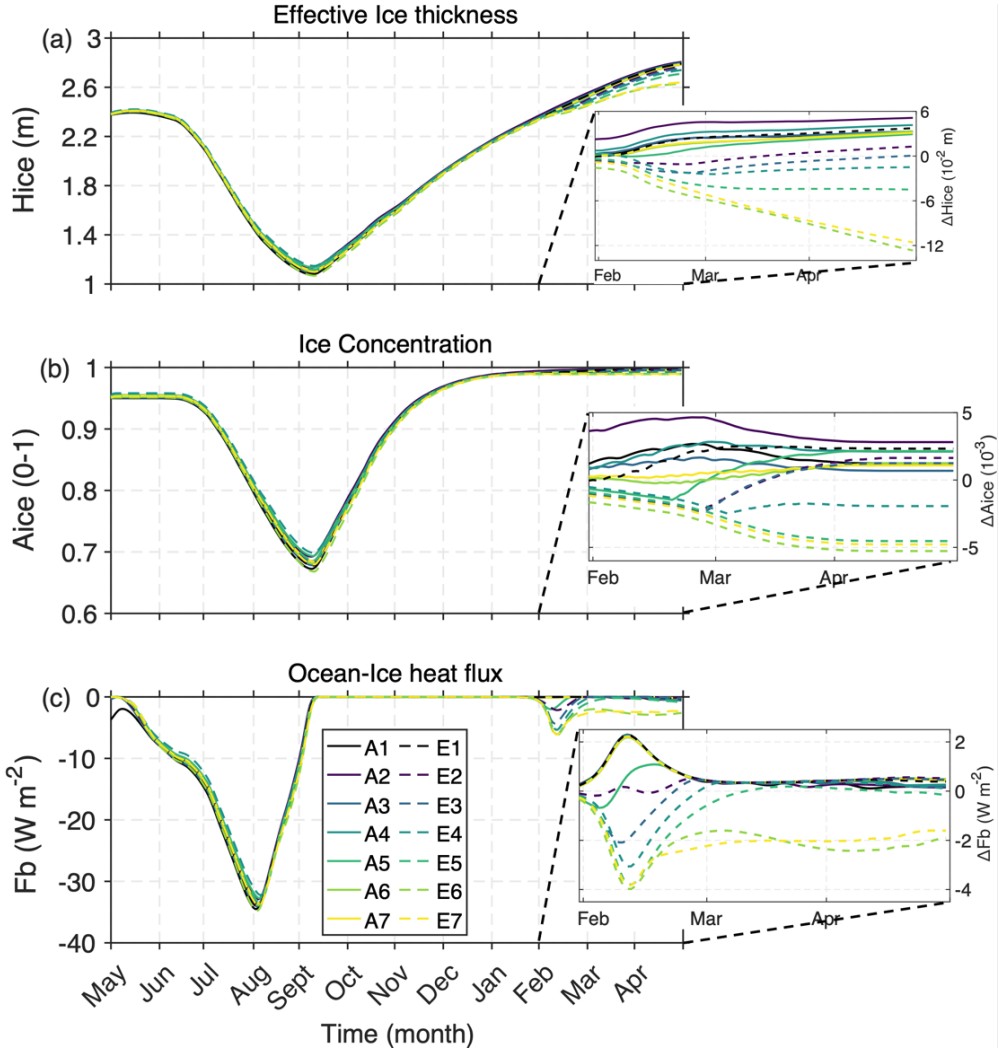

**Figure 6: Time series of the (a) effective sea ice thickness (Hice), (b) ice concentration (Aice) and (c) ocean-ice heat flux ($F_b$, negative values representing the heat transfer from the ocean to the ice) for all control runs. The amplified subplot shows the anomalies (each control run minus the average of all control runs) during the months of February to April.**

In the control run, the calculated ocean-ice heat flux in the Canada Basin (stations A1-A7) has an average value of 0.06W m$^{-2}$ during the freezing season and 15.8 W m$^{-2}$ during the melting season (Fig. 6c). The observed ocean-ice heat flux from the entire surface heat budget of the Arctic Ocean drift has an average value of 2.2 W m$^{-2}$ during winter and 16.3 W m$^{-2}$ during summer (Shaw et al., 2009). This comparison indicates that the simulated ocean-ice heat flux is close to the observations in

summer but much smaller than the observations in winter. The main reason for this is the omission of horizontal advection in
the one-dimensional model. Horizontal heat transport is an important factor in the increase in ocean-ice heat flux in winter
and omitting it from the model will lead to a lower winter ocean-ice heat flux. In summer, the main heat source is the absorption
of solar radiation (Perovich et al., 2011) and the surface water temperature is less affected by horizontal heat advection.

As observed by Jackson et al. (2010) and Steele et al. (2011), our model also shows that the NSTM normally deepens, cools,
and disappears throughout the autumn and winter (Fig. 4). However, it has been observed as a year-round feature sometimes.
Jackson et al. (2012) found that when ITP18 drifted into shallow waters from early to mid-December, the ocean-ice heat flux
reached up to 55 W m$^{-2}$ (Jackson et al., 2012), reduced sea ice thickness at the end of the 2008 growth season by about 25%
(Timmermans, 2015). Smith et al. (2018) also discovered occasional high values of the winter ocean-ice heat flux (about 100
W/m$^2$) in the Canadian Basin using ITP and CTD data from 2015. These high winter sea-ice heat fluxes are usually
associated with strong wind events (Smith et al., 2018). In this study, all experiments utilized regionally averaged wind fields
to eliminate the impact of wind field variability. This may be the reason why our one-dimensional model did not reproduce
the episodic high values of the ocean-ice heat flux in winter successfully.

Simulated ocean-ice heat fluxes in the Amundsen Basin (station E1-E5) and Nansen Basin (station E6-E7) have an average
value of 0.29 W m$^{-2}$ and 1.2 W m$^{-2}$ during the freezing season and 15.6 W m$^{-2}$ and 16.1 W m$^{-2}$ during the melting season (Fig.
6c), respectively. The ocean ice heat flux in the Eurasian Basin in winter is larger than that in the American Basin, which
results in less ice formation in the Eurasian Basin. The results suggest that ocean stratification is a very important factor for
ice growth, which agrees with the conclusions of Linders and Björk (2013).

### 3.2 Melt water perturbation experiments

### 3.2.1 Upper Ocean responses

### a) Summertime

Figure 7 shows the temperature and salinity profiles in summer for the MWP and control runs. It is obvious that no release of
melt water has the most pronounced effects, compared to the control run, while the release a portion of melt water has moderate
to little effect on the upper ocean structure. The experimental results for some stations in this study are very similar, so this
paper shows the simulation results for six representative stations to show the general behavior of the model and the impact of
ocean stratification. Three of them are located in the Amerasian Basin (A2, A4 and A6) and three in the Eurasian Basin (E2,
E6 and E7).

In the MWP-0% runs, as no melt water is released, the upper water is swell mixed, the NSTM vanishes, and the temperature
and salinity are uniform in the upper layer (down to a depth of several tens of meters) in the Canada Basin to more than 100
m depth in the Nansen Basin (Fig. 7). As a result of mixing, compared with the control run, salinity increases, and temperature

decreases in the ML at stations A1-A7 (such as Fig. 7a-c) and E1-E5 (such as Fig. 7d). However, at stations E6 and E7, the temperature decreases in the upper ML but increases in the lower ML (Fig. 7e, f). The strength of ocean stratification at stations E6 and E7 are very weak, and the removal of all the melt water leads to downward transfer of heat stored in the NSTM, which warms the lower ML.

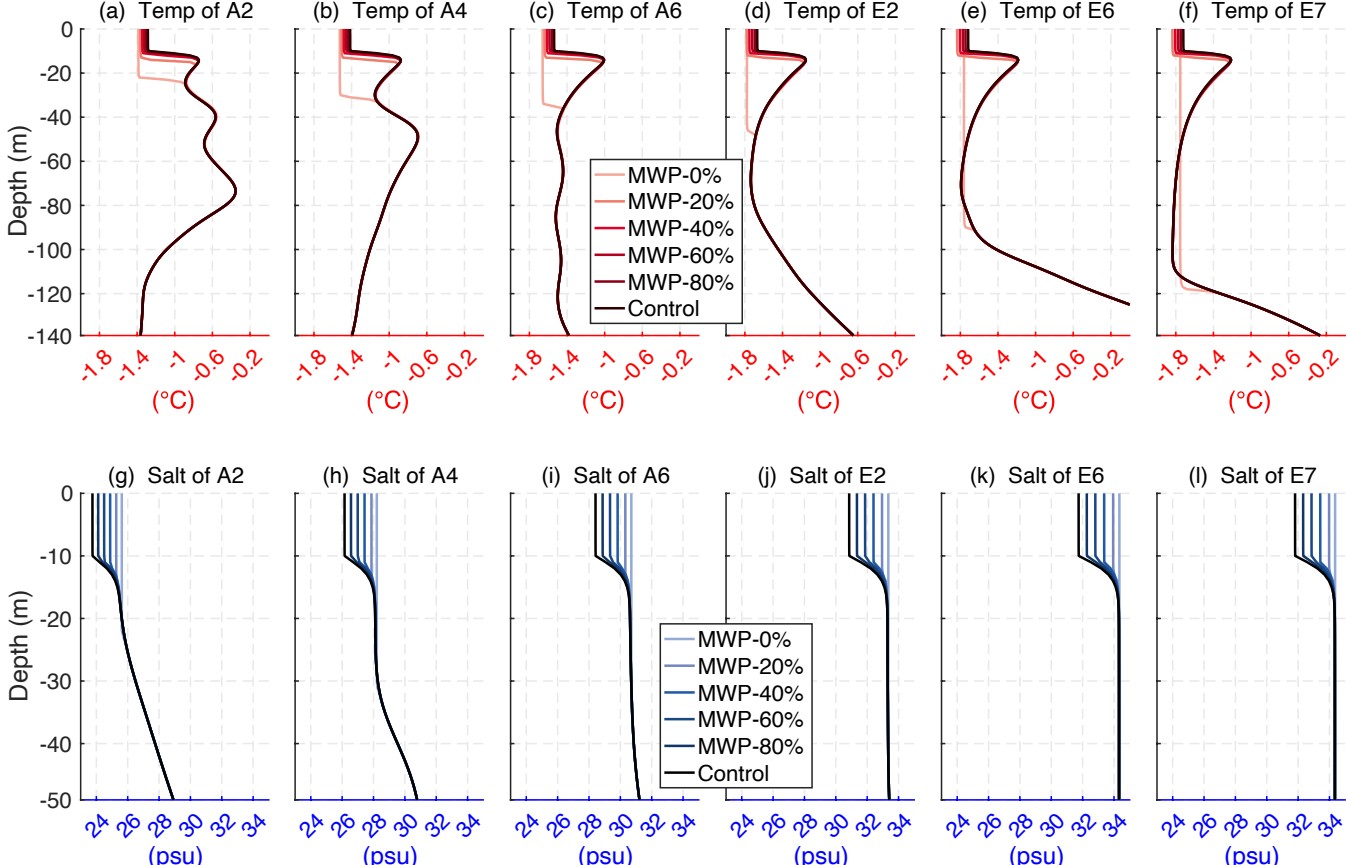

**Figure 7: Simulated temperature (top row) and salinity (bottom row) profiles of control runs and MWP runs in mid-August for stations A2, A4, A6, E2, E6 and E7.**

In contrast to the MWP-0% run, the summer MLD in the MWP 20%-80% runs are no more than 10m, which implies that a certain amount of melt water is sufficient to maintain upper ocean stratification during summer. When all the melt water is removed from the model, the MLD can reach 22-44 m in the Americana Basin, 33-90 m in the Amundsen Basin and over 100 m in the weaker stratified Nansen Basin at the end of the melting season (Fig. 8).

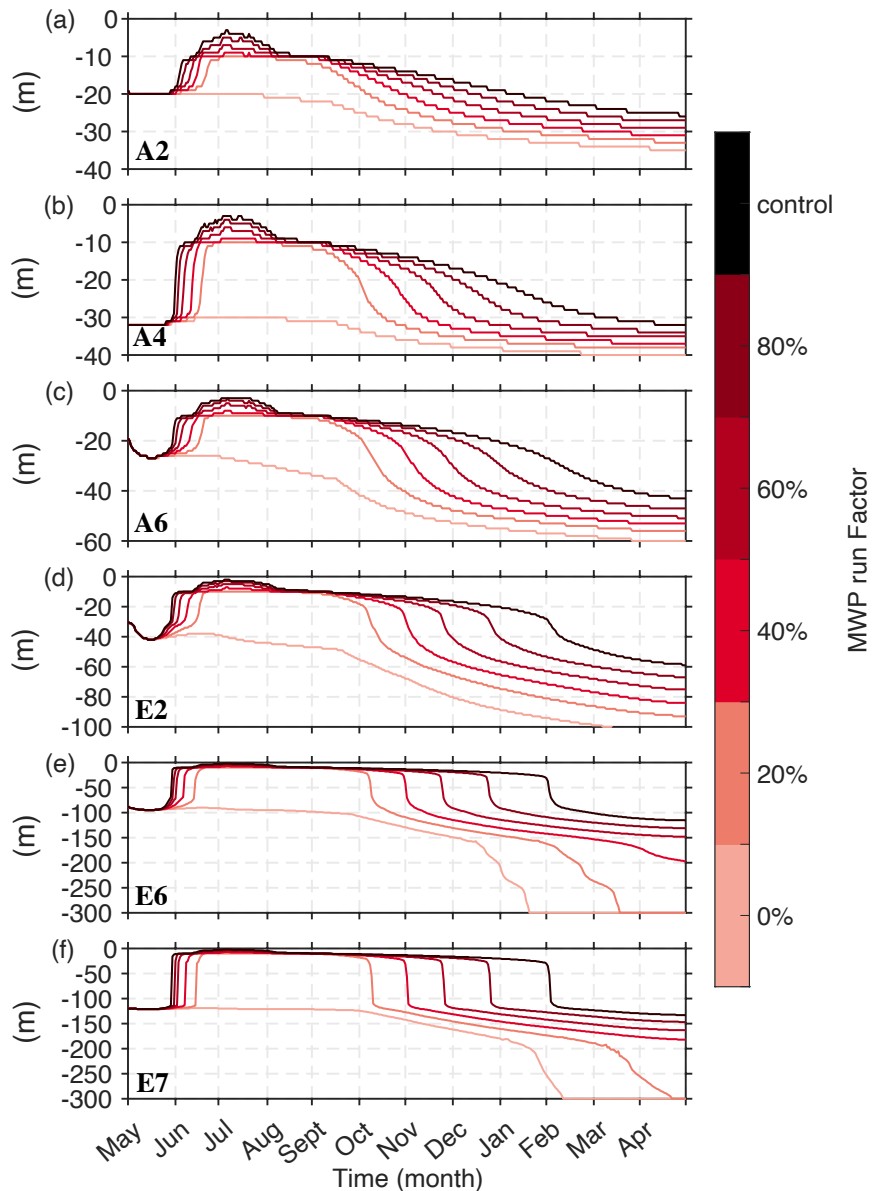


**Figure 8: Time series of the MLD of the control and MWP runs for stations (a) A2, (b)A4, (c) A6, (d) E2, (e) E6 and (f) E7. The color of each line represents the MWP run factor.**

**b) Wintertime**

Figure 9 shows the temperature and salinity profiles in winter for the MWP and control runs. The extent to which melt water
affects the ocean profile varies with stations (Fig. 9). At each station, the MLD increases following the reduction in the release
of melt water in the previous melting season. Close to the end of the freezing season (mid-April), the MLD reaches its

maximum at all stations. At stations A1-A4 in the Canada Basin, the MLD is 35-44 m for the MWP-0% run and is still unable to penetrate the PSW layer (Fig. 8a, b and Fig. 9a, b). The MLD in the Amundsen Basin is much larger than that in the Canada Basin in the MWP-0% runs, approximately 42-170 m (33-99 m in the control run) (Fig. 8d). Nevertheless, it is still unable to reach the core of the warm AW (Fig. 9d).

Stations E6 and E7 in the Nansen Basin show a relatively extreme situation in the 20% and 0% runs during winter. The removal more than 20% of the melt water leads to the ML dropping to a depth of more than 300 m (116 m and 133 m in the control run, respectively), which can reach the core depth of the warm AW (Figs. 8e and f). This led to a dramatic change in the structure of the vertical profile when the AW layer was well mixed with the cold water in the upper layers (Fig. 9e, f, k and l). The heat carried by the warm AW will melt the surface ice and release significant amounts of heat into the atmosphere, as described in the next section. The results suggest that the positive buoyancy flux of the melt water is a significant impediment to the deepening of the ML throughout the simulation.

The above results of the MWP runs imply that the subsurface PSW in the Canada Basin is unable to reach the ice even when all the melt water is removed due to strong stratification. However, at some places in the Nansen Basin, such as at stations E6 and E7, that lack a fully developed halocline, melt water plays an important role in preventing the ML from reaching the AW layer.

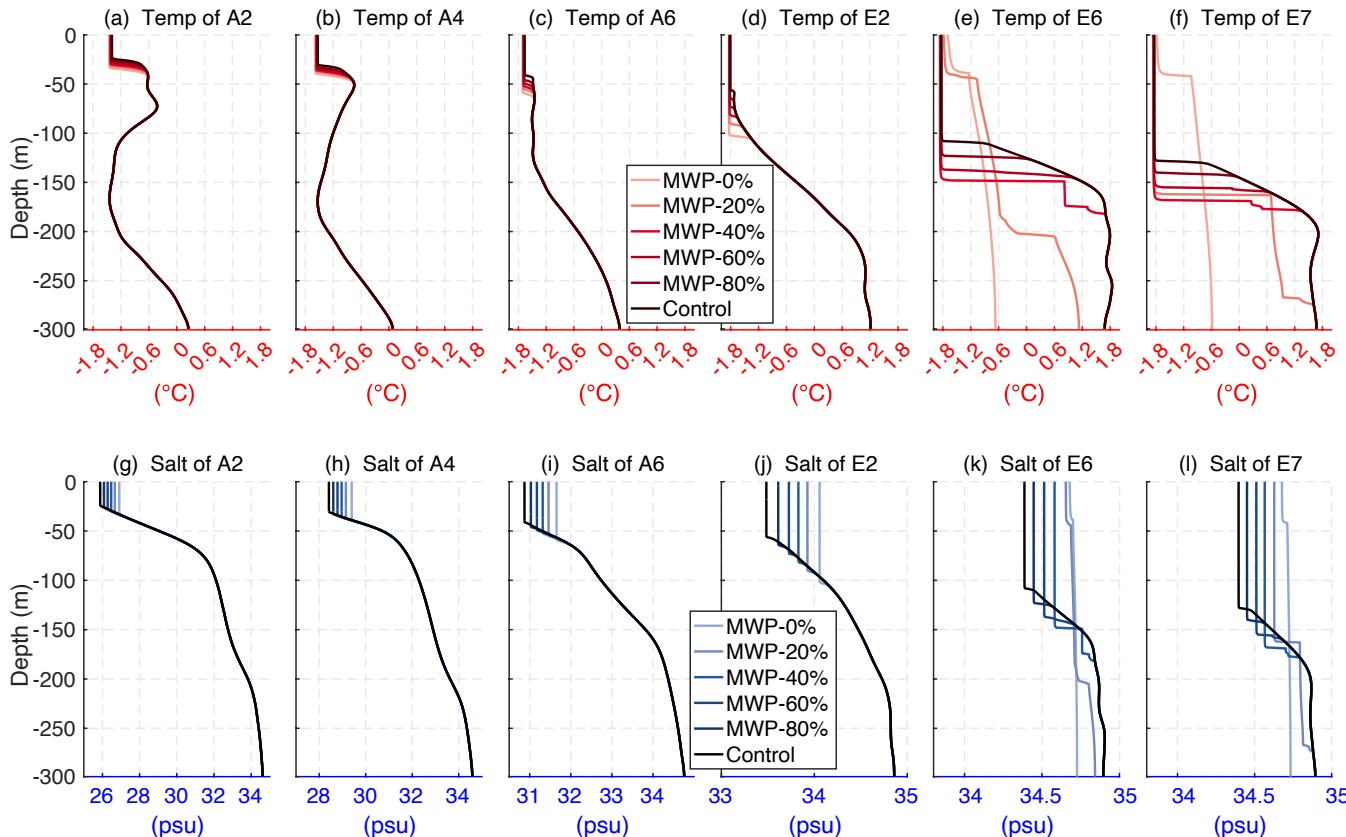

**Figure 9: Same as figure 7 but in mid-April.**

### 3.2.2 Sea ice responses

#### a) Melting season

The reduced melt water release leads to decreases in the summertime effective ice thickness and ice concentration (Fig. 10). In comparison to the control run, the amount of melting ice increases by 21.6 cm (~16.6%), 6.4 cm (~4.9%), 3.8 cm (~2.9%), 2.4 cm (~1.8%) and 1.2 cm (~0.9%) (averages of all stations) for the MWP-0%, 20%, 40%, 60% and 80% runs, respectively, over the entire melting season (Fig. 11a). This suggests that the removal of melt water promotes ice melting, i.e., melt water has negative feedback on sea ice melting during summer.

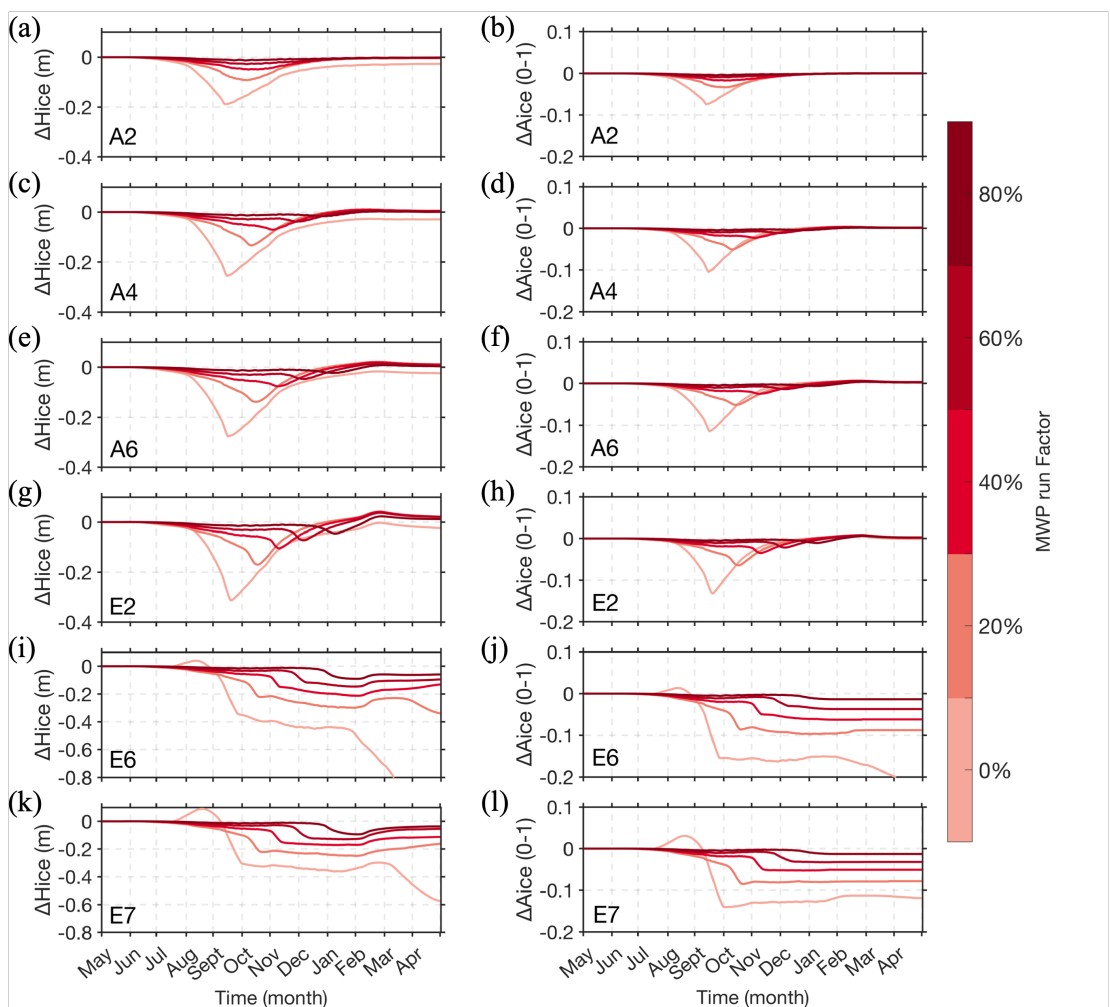

**Figure 10: Time series of (left) the anomalies of effective ice thickness and (right) anomalies of ice concentration for stations A2, A4, A6, E2, E6 and E7. The anomalies are obtained from the MWP run minus the control run.**

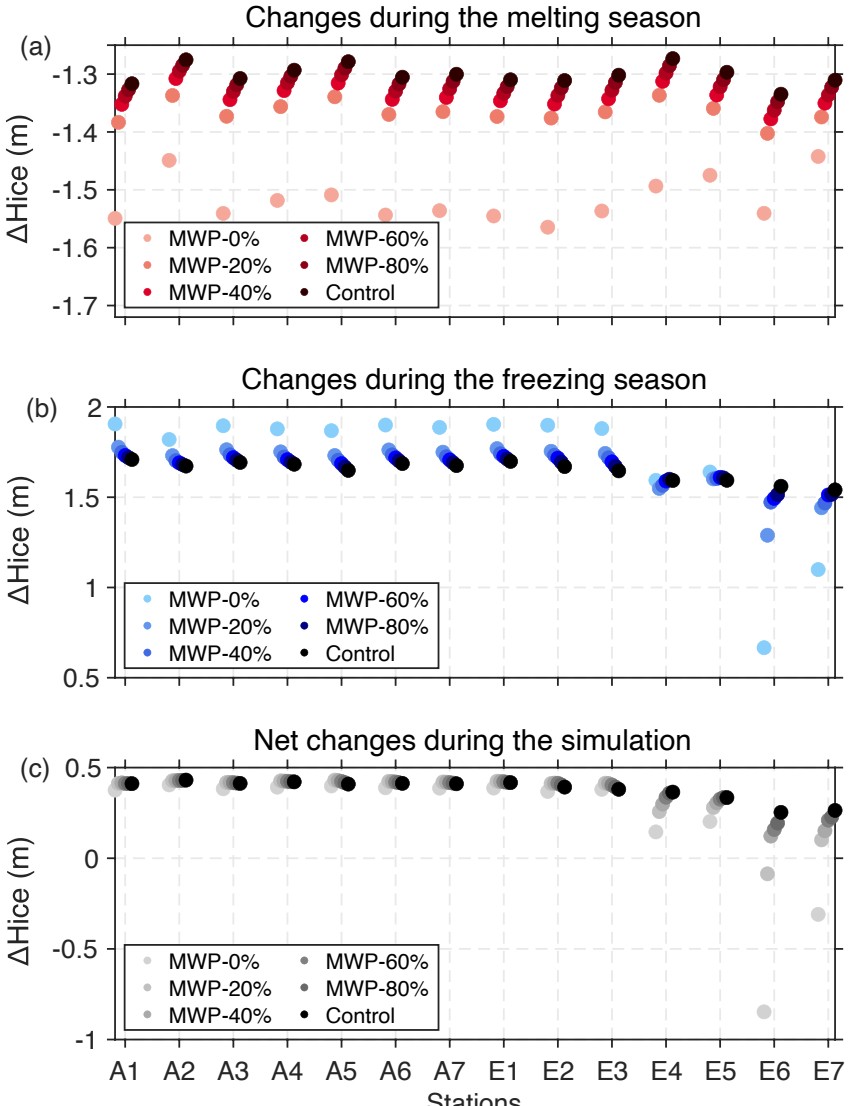

**Figure 11: Ice thickness change during model simulation for all stations. (a) Effective ice thickness change during the melting season. (b) Effective ice thickness change during the freezing season. (c) Difference in effective ice thickness between the end value and the initial value.**

The time series of the ice top and bottom change rate anomaly between the MWP and the control runs are shown in Fig. 12. The melt water feedback acts mainly on the ice bottom, as opposed to on the ice top. The melt water affects bottom melting mainly by impeding vertical mixing of the heat stored in the subsurface. In the MWP-20% to 80% runs, as the melt water release decreases, the summer halocline weakens, allowing more heat in the NSTM to mix upwards, resulting in a larger ocean-ice heat flux (Fig. 13, left column).

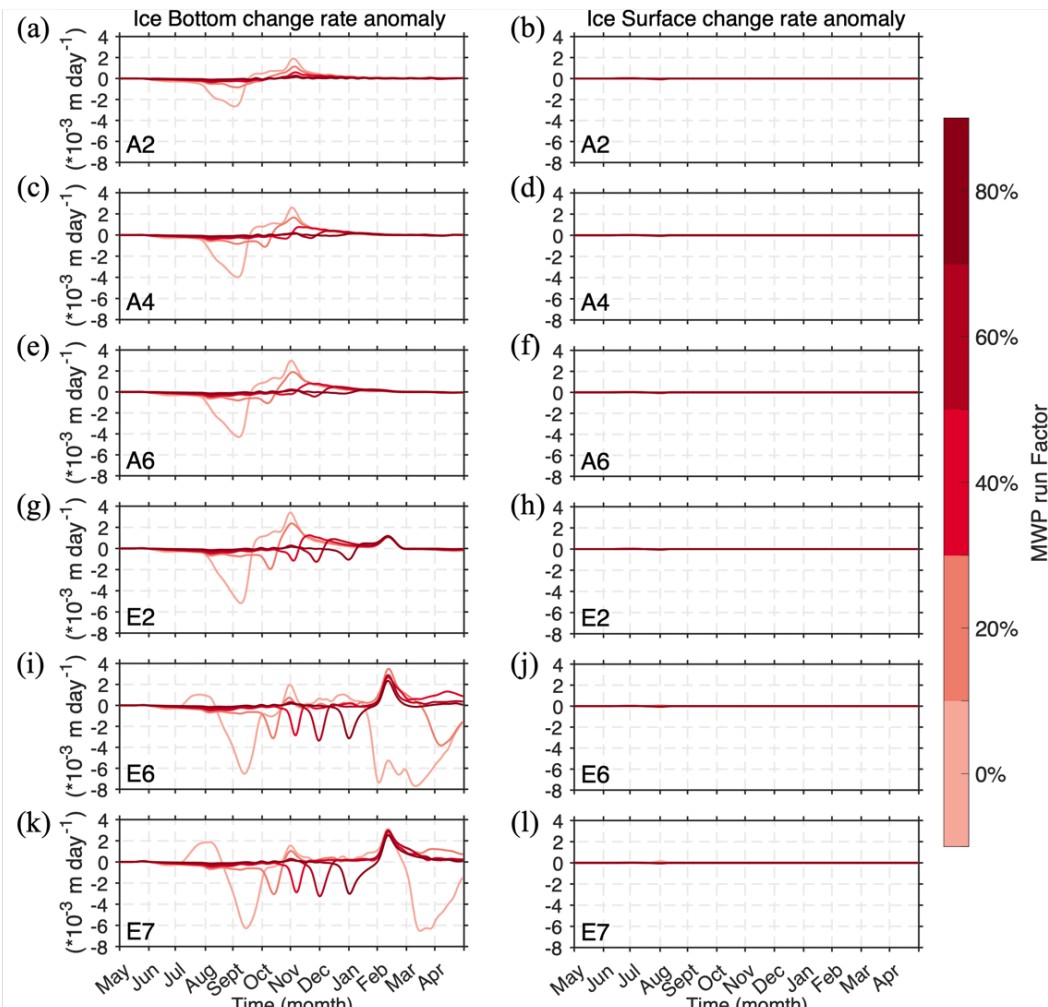

Figure 12: Time series of (left) the anomalies of ice bottom change rate and (right) the anomalies of ice surface change rate for stations A2, A4, A6, E2, E6 and E7. The anomalies are obtained from the MWP run minus the control run. The negative (positive) values indicate faster (slower) rates of ice decrease in the MWP run compared to the control run. The color of each line represents the MWP run factor.

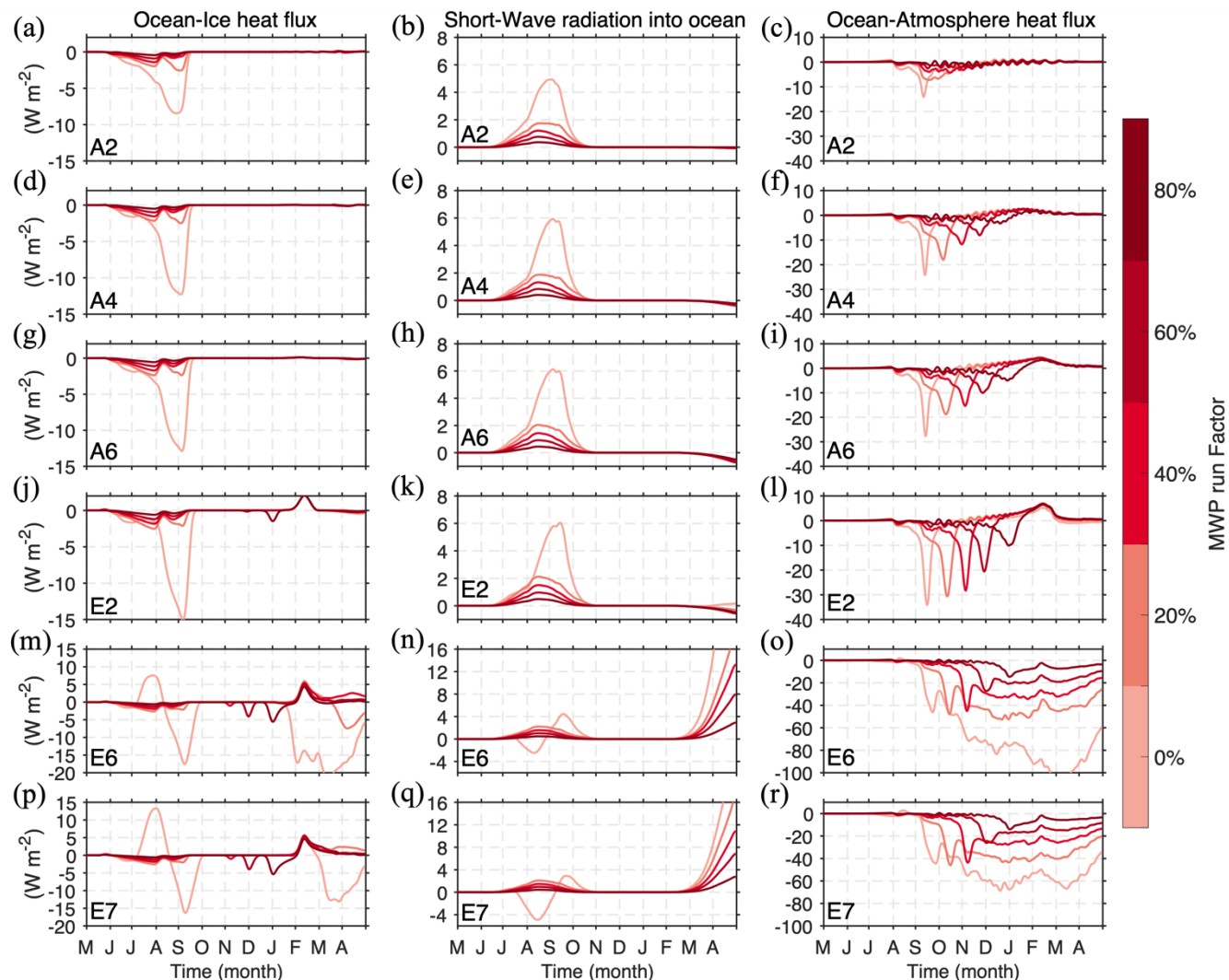

Figure 13: Time series of (left) the anomalies of ocean-ice heat flux, (middle) the anomalies of shortwave radiation, and (right) the anomalies of ocean-atmosphere heat flux for stations A2, A4, A6, E2, E6 and E7. The anomalies are obtained from the MWP run minus the control run. The negative (positive) value indicates heat gain (loss) by the ocean in the MWP run compared to the control run. The color of each line represents the MWP run factor.

In the MWP-0% runs, the NSTM promotes ice bottom melting in two ways. The first way, which is dominant in well-stratified areas, is by directly heating the ice bottom by upwards mixing in summer, resulting in faster melting, and the other way, which is dominant in areas with weaker stratification, is by prolonging the melting season. For example, at station A1, where the stratification is strong, the ice bottom melting rate (Fig. 12a) and ocean ice heat flux (Fig. 13a) are greater in the MWP-0% run than the control run throughout the whole summer, while at stations E6 and E7, where stratification is weak, the ice bottom melting rate (Fig. 12k) and ocean-ice heat flux (Fig. 13p) are not greater in the MWP-0% run than the control run until late

summer. The reason is that in the strongly stratified stations, even when all melt water is removed, the stratification is still strong, and the heat stored in the NSTM is mixed only upwards and used for ice melting, whereas at a weakly stratified station, the heat is not only mixed upwards but also mixed downwards. The heat transferred downwards to the lower ML mixes upwards at the onset of the freezing season, which delays the freeze-up and prolongs the melting season.

**b) Freezing season**

Figure 11b shows the effective sea ice thickness changes from the minimum value in summer to the end of the freezing season in the sensitivity experiments for all stations. The winter sea ice formation at the strongly stratified stations A1–A7 and E1-E3 are inversely proportional to the amount of melt water released in previous melting season. In the MWP-0% run, an average increase in sea ice thickness of 20.6 cm (approximately 12.3%) was simulated at these stations compared to the control run. Sea ice formation at stations E4 and E5 is less sensitive to melt water release changes than that at other stations. In contrast, at

the weakly stratified stations E6 and E7, sea ice formation in the MWP-0% runs decreases by an average of 66.8 cm (43%) compared to the control run (Fig. 11b).

At some stations with strong haloclines, e.g., stations A1-A7 and E1-E3, even with all the melt water removed, the halocline is still strong, which can effectively prevent the ML from deepening in autumn and winter. In particular, the reduction in summer melt water leads to a weakening or even absent NSTM, and that there isn't enough heat stored in the subsurface layer

to replenish the heat loss at the surface when autumn arrives, leading to a more rapid cooling of water temperature to the freezing point in autumn, and hence increasing ice formation in autumn. This result suggests that the presence of the NSTM effectively hinders sea ice growth in autumn, which is consistent with Toole et al. (2010)'s results. However, at some stations with a weak halocline, e.g., stations E6 and E7, the Atlantic warm water can reach the surface, which effectively prevents the formation of sea ice. In addition, in March, the ML can reach the depth of the warm AW, and a large amount of heat from the

warm AW mixes upwards and heats the sea ice, leading to early melting of the sea ice (such as Fig. 10i, k), which allows large areas of open water to exist during the winter (such as Fig. 10j, l), thus enabling the sea surface to absorb more solar radiation in April (Fig. 13n and q), allowing heat from the warm AW to enter the atmosphere, and the ocean-atmosphere heat flux can reach 70-100 W/m$^2$ in March at stations E6 and E7 (Fig. 13o and r).

The results indicate that the impact of melt water released during the previous melting season on winter sea ice growth depends

on the strength of stratification, with gradually transitions from promoting to impeding ice growth as the halocline weakens.

**c)  Annual net sea ice changes**

The annual net changes in the effective ice thickness for the control run and MWP runs at all stations are shown in Fig. 11c. In strongly stratified regions (such as stations A1-A7 and E1-E3), the annual net sea ice change is insensitive to melt water release (Fig. 11c) because the reduction in melt water not only leads to more sea ice melting in summer but also leads to an

380 increase in ice formation during winter (Fig. 11b), which offsets the extra ice melting in summer. In weakly stratified regions (stations E4-E7), the annual net sea ice change is more sensitive to melt water release (Fig. 11c) because the reduction in melt water induces a deeper ML and enhances the ocean-ice heat flux, resulting in insufficient sea ice formation in winter, which cannot compensate for the extra summer ice melting.

In summary, the above results indicate that melt water always has negative feedback on ice melting during melting season. 385 The impact of melt water released during the previous melting season on the subsequent winter ice formation depends on the strength of stratification. It hinders (promotes) ice formation in areas with strong (weak) stratification. The presence of the melt water hinders the transfer of heat from the subsurface to the ice cover, which is the main reason for the negative feedback of melt water on sea ice melting during summer. In addition, the melt water significantly inhibits ice melt at stations E6 and E7 by hindering the upwards heat flux from warm AW in spring.

390 **3.3 Sensitivity experiments with thinner sea ice**

In recent decades, it has been observed that Arctic summer sea ice appears to be decreasing rapidly (Perovich et al., 2019), with larger ice-free areas in summer and thinner winter sea ice (Haine and Martin, 2017). Thus, several experiments are conducted using thinner initial ice (1.5 m). To highlight the effects of strong or weak CHL, we selected stations A3, A6, E2 and E7 to do the thinner ice experiments.

395 In the control run, the initial thinner ice of 1.5m completely melts in late July (Fig. 14a), and the maximum ocean-ice heat flux can reach 330Wm$^{-2}$ (Fig. 14b). During winter, E7 station produces less sea ice because it possesses a weaker stratification (see Fig. 14a), which is consistent with experiments that had an initial ice thickness of 2.5 m.

Compared to the control runs and the MWP20%-80% runs, the sea ice melts more slowly in the MWP-0% runs (Figures 14c-f), which contrasts with the experiments with a thicker initial ice. This may be due to the fact that the thinner initial ice 400 contribute to the presence of a larger open ocean during the summer and increased wind input enhances the mixing level, resulting in more heat being mixed into the deeper ocean. As a result, the heat available for melting sea ice is reduced. Figures 15a-d clearly demonstrate the process by late July, the temperature of the upper ocean is remarkably lower in the MWP-0% runs, while the temperature below 10m is considerably higher compared to the other runs.

During winter, the role of melt water in hindering the upward mixing of AW is more evident in the thinner initial ice 405 experiments. Removing 40% of melt water during the summer in the thinner initial ice runs can enable the upward mixing of the AW (Fig. 16d and h) and subsequent melting of sea ice in winter (Fig. 14f and Fig. 17b). However, it would require the thicker initial ice runs to remove over 80% of melt water to achieve similar results (Fig. 9f and l).

The thinner ice experiments indicate that as multi-year ice in the Arctic Ocean is replaced gradually by seasonal sea ice, melt water will play a more significant role in impeding vertical mixing and winter ice melting in the future.

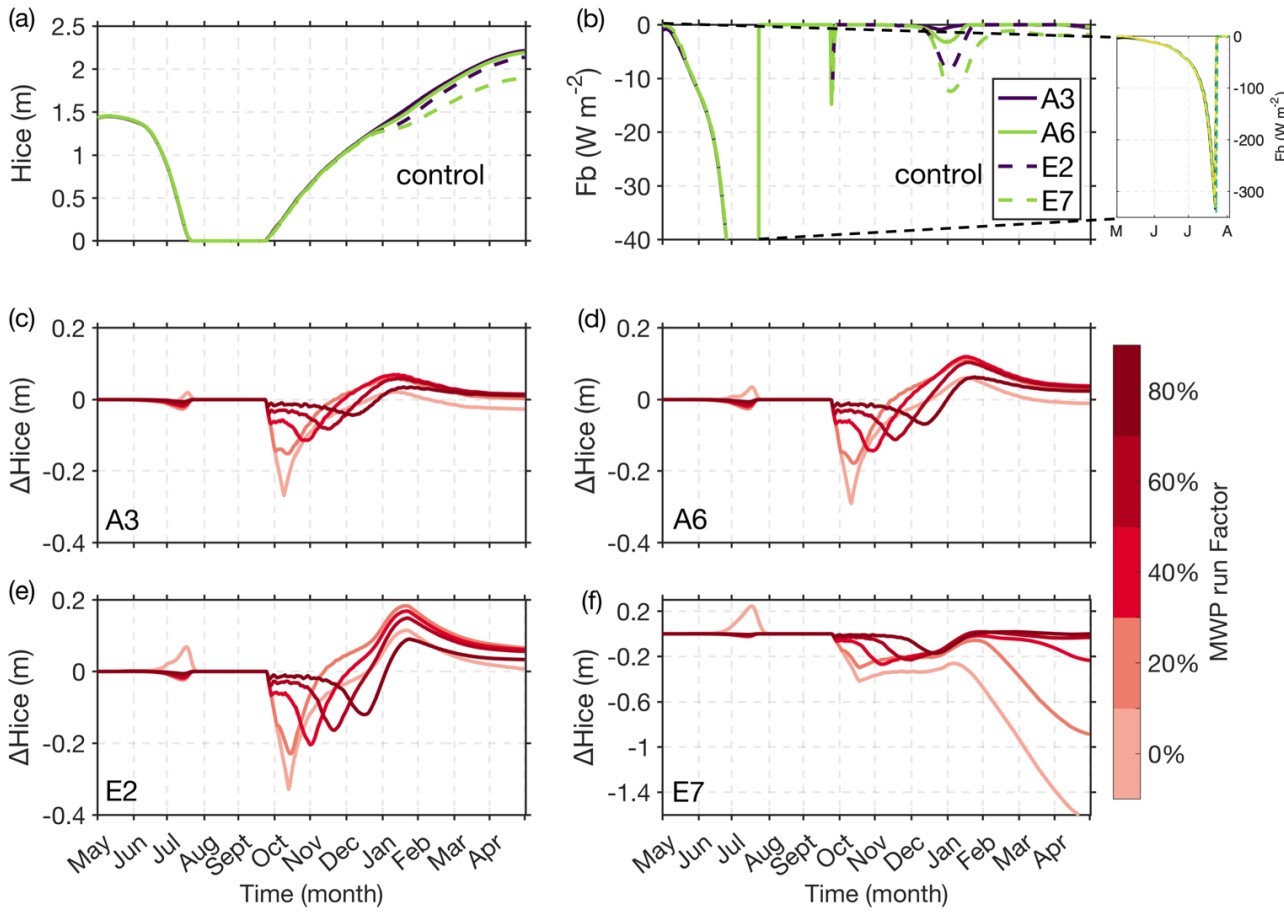


**Figure 14: Time series of the (a) effective sea ice thickness and (b) ocean-ice heat flux (negative values represent the heat transfer from ocean to ice) for control runs with thinner initial ice thickness. The subplot in (b) shows the time series of ocean-ice heat fluxes between May and August, indicating that ocean-ice heat fluxes can reach a maximum of 330Wm⁻². (c)-(f): Time series of the anomalies of effective ice thickness for stations A3, A6, E2 and E7. The anomalies**

**are obtained from the MWP run minus the control run.**

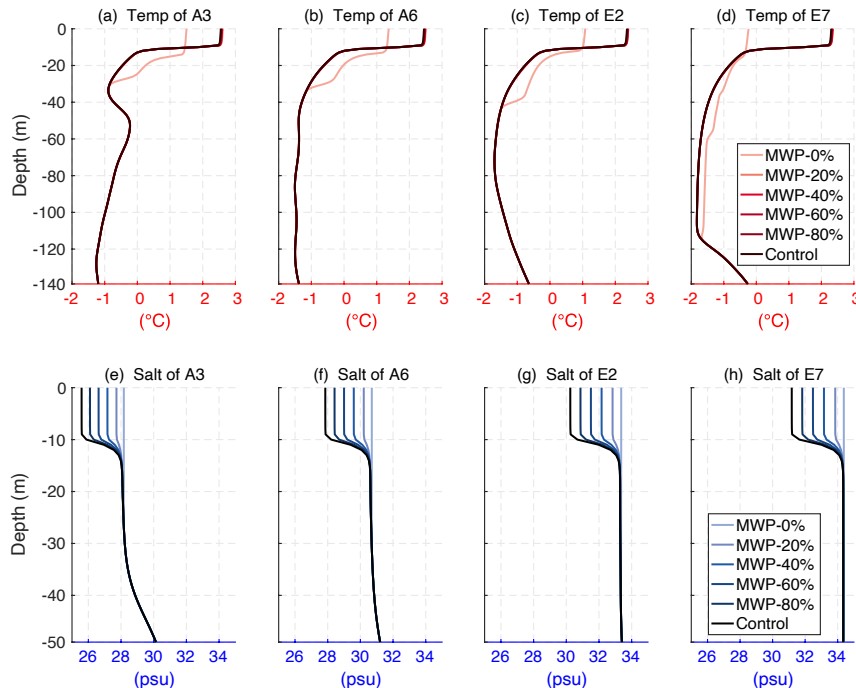

**Figure 15: Simulated temperature (top row) and salinity (bottom row) profiles of control runs and MWP runs in late-July for stations A3, A6, E2, and E7 of the thinner initial ice experiments.**

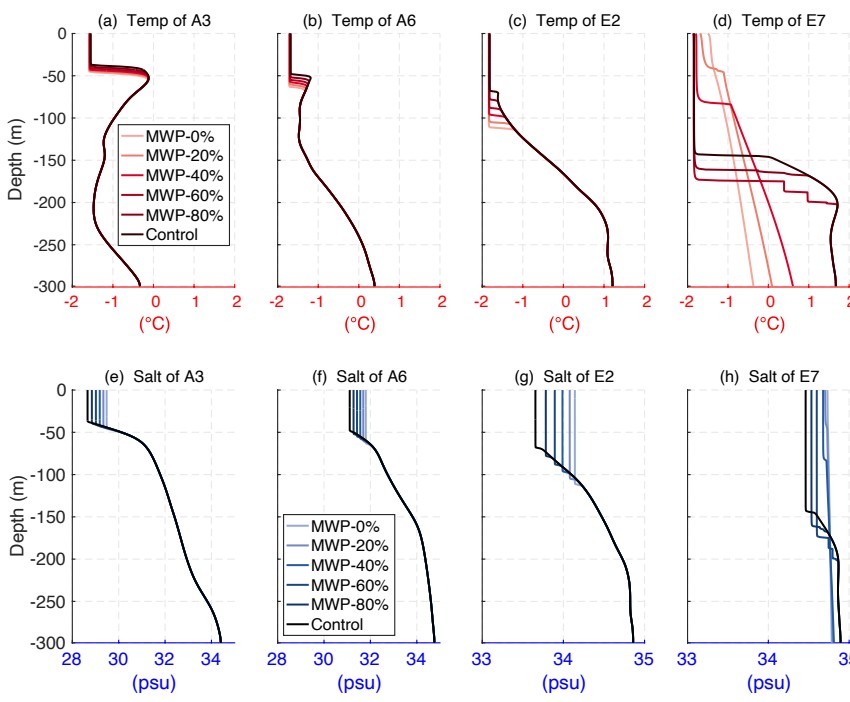

420 **Figure 16: Same as figure 15 but in mid-April.**

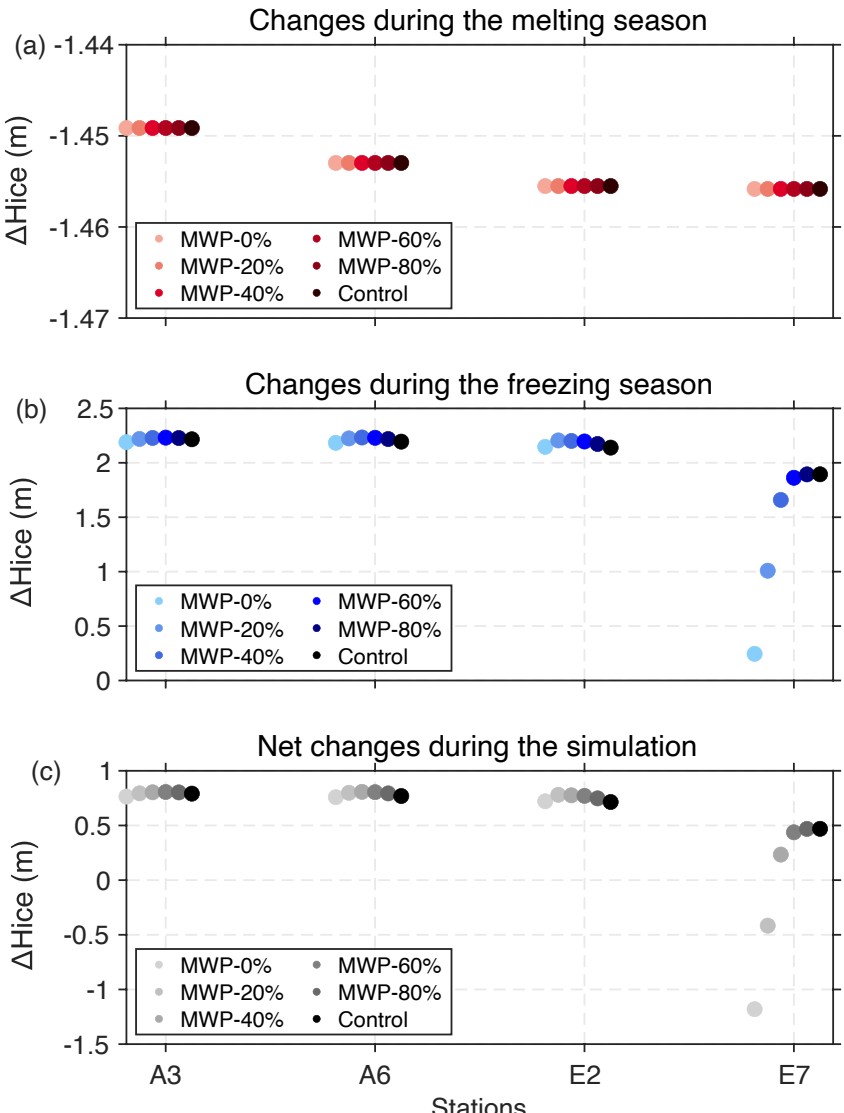

Figure 17: Same as figure 11 but for stations A3, A6, E2 and E7 of the thinner initial ice experiments.

## 4 Discussion

A key finding of this study is that the impact of melt water on sea ice varies based on the strength of ocean stratification. This

suggests that the ice-covered Arctic Ocean can be divided into two regimes based on ocean stratification: some areas with strong stratification that are less sensitive to melt water, where the melt water only prevents the NSTM from melting ice, and other small areas with weak stratification that are more sensitive to melt water, where the melt water not only prevents NSTM from melting ice in the summer but also prevents warm AW from mixing upwards in early spring. The border between these two regimes depends on ocean stratification. Stations E4 and E6 are clear examples. The initial CHL at station E6 is quite

weak (Fig. 2f), and the AW is mixed sufficiently upwards in the spring after removing the melt water. Station E4, close to station E6, has a relatively stronger initial CHL than station E6 (Fig. 2f), and the AW cannot reach the upper ocean and ice even when the melt water is removed.

Wang et al. (2019) noted that although sea ice decline does not change the total Arctic liquid freshwater content (FWC), the increase in the liquid FWC in the Amerasian basin is nearly compensated by the reduction in the Eurasian basin, which results
in significant changes in the spatial distribution of the liquid FWC. This raises the question of how the exchange of melt water between these two regimes in the Arctic Ocean will affect the ocean and sea ice. From our experimental results, it appears that if melt water or liquid FWC from other sources is continuously lost outwards from an area during the melting season, then the sea ice in that area will melt more rapidly; if this area occurs where the melt water will have a significant impact, then there is a high probability that the AW will mix sufficiently upwards during the freezing season and reach the ice cover, which can
cause substantial sea ice melting and prolong the melting season. The warm AW plays an important role in reducing the sea ice cover in the Arctic Ocean through upwards heat loss (Polyakov et al., 2010) because the heat contained within the AW layer is sufficient to melt all sea ice in the Arctic within a few years (Turner, 2010). Climate model projections suggest that freshwater input from enhanced river runoff and positive precipitation minus evaporation (P-E) will increase by ~30% by 2050 (Peterson et al., 2002; Bintanja and Selten, 2014; Haine et al., 2015). Increased freshwater input, like more melt water entering
the ocean, can strengthen the cold halocline by increasing the magnitude of the salinity gradient, which will also inevitably have an impact on sea ice melt/production, especially in some important areas such as stations E6 and E7 in this study.

A limitation of the one-dimensional model is that it cannot directly represent the effect of lateral variations in the upper ocean in combination with ocean/ice advection. But in this study, we focus on the effects of melt water on vertical processes in the ocean and do not consider the effects of advection, and the simulations are short (one year). For shorter simulations and
relatively horizontally constant temperature and salinity properties, it can be assumed that advection will have a smaller effect (Linders and Björk, 2013). Therefore, the results of the one-dimensional mode used in this study can be justified. Advection is of great importance when performing long-term simulations and should be addressed, for example, by introducing some type of restoration of the profiles towards the observed values (Polyakov et al., 2010, 2013b). In addition, as mentioned in section 3.1.2, changes in wind speed will affect stratification and the melt/formation of sea ice through increased vertical
mixing. The ideal modelling method used in this study cannot reproduce the episodic high values of ocean-ice heat flux caused by wind mixing, as reported by Jackson et al., (2012) and Smith et al., (2018). Therefore, it is necessary to consider the wind speed on the role of melt water in the sea ice-ocean coupled system in future work.

This study provides valuable insights into the intricate relationships between ocean stratification, melt water, and sea ice growth and their implications for predicting future changes in the Arctic region. Understanding these complex interactions is essential
for developing accurate climate models and assessing the potential impacts of climate change on the Arctic ecosystem. The study in this paper addresses only the effects of melt water in the vertical direction, and future work could focus on the effects

of melt water transport processes during the melting season in conjunction with Arctic Ocean circulation. To address this issue, more detailed modelling, including advection processes, is needed.

## 5 Conclusions

In this study, the responses of upper ocean stratification and sea ice melt/formation in the Arctic Ocean to melt water release are investigated using a one-dimensional coupled sea ice-ocean model. We perform two types of experiments to achieve the goals: a control run and five melt water perturbation experiments with 0%, 20%, 40%, 60%, and 80% melt water release into the ocean.

Compared to the observations, the one-dimensional coupled sea ice-ocean model reproduces the observed temperature and
salinity structure of the Arctic Ocean reasonably well, capturing important features such as the fresh surface layer, the near-surface temperature maximum (NSTM), and the seasonal variation in MLD. In the control runs, the results suggest that ice growth depends on ocean stratification because weaker ocean stratification leads to higher ocean-ice heat flux during winter. In the melt water perturbation experiments, as expected, decreasing melt water increases the salinity of the surface and weakens stratification, flattening the upper halocline and changing the vertical heat flux from the depth to the surface. These changes
subsequently affect the melting or formation of sea ice. Our results suggest that a decrease in melt water release has the following effects on sea ice:

    1. During the melting season, melt water has an inhibitory effect on sea ice melt by preventing upward mixing of heat from the subsurface layer. The minimum summer effective sea ice thickness values in the control runs are approximately 16.6% greater than those of the MWP-0% runs, suggesting that the presence of melt water exerts
negative feedback on the process of sea ice melt.

    2. During the freezing season, the effect of melt water released in the previous melting season on sea ice growth varies with ocean stratification. In regions with weaker stratification, such as the Nansen Basin, melt water plays a more important role in maintaining sea ice and ocean stratification than in areas with stronger stratification, such as the Canadian Basin. The model results show that at strongly stratified stations, the net increase in winter effective sea ice
thickness in the control run is approximately 12.3% smaller than that in the MWP-0% runs. Conversely, at weakly stratified stations, the net increase in effective sea ice thickness in the control run is approximately 43% larger than that in the MWP-0% runs. Our findings reveal that the effects of melt water from the previous melting season on the subsequent winter ice formation depend on the strength of stratification. Specifically, it impedes ice formation in areas with strong stratification, while it promotes it in areas with weak stratification.

3. Sensitivity experiments with thinner initial ice indicate that as multi-year ice in the Arctic Ocean is gradually replaced by seasonal sea ice, melt water will play a more significant role in hindering vertical mixing and winter ice melt in the future.

## Data availability

The Ice-Tethered Profiler data are collected and made available by the Ice-Tethered Profiler Program (Krishfield et al., 2008;
Toole et al., 2011) based at the Woods Hole Oceanographic Institution (https://www2.whoi.edu/site/itp/). NCEP/DOE
Reanalysis II data provided by the NOAA PSL, Boulder, Colorado, USA (https://psl.noaa.gov). The numerical model
configuration, parameters, forcing fields and the simulation results used in this paper are stored at
https://zenodo.org/record/7727849#.ZA7nuOtBxTY.

## Author contributions

HH designed and conducted the experiments, analyzed the experimental data, and drafted the initial version of the manuscript.
XZ conceived the idea for the study, participated in writing the manuscript, and made several significant revisions to it. KW
contributed to the analysis of experimental data. All authors have reviewed and approved the final version of the manuscript.

## Competing interests

Neither of the authors has any competing interests.

## Acknowledgements


This work was funded by the National Key Research and Development Program of China (Grant 2017YFA0604602), the
National Natural Science Foundation of China (Grant 42276254), and the Postgraduate Research & Practice Innovation
Program of Jiangsu Province (Grant KYCX19_0384). We thank the two reviewers for their constructive comments and
suggestions.

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
