# Peer review of "A numerical study on melt water feedback in the coupled Arctic Sea ice-ocean system"

_EGUsphere, 2023_

## Author Comment (AC2)

**Responses to Reviewer #1**

**The manuscript by Zhang et al. uses a 1-D coupled sea ice-ocean model to examine the impacts of melt water on sea ice melt and upper ocean stratification. Sensitivity analyses were run to compare how the percent of melt water changes the upper ocean stratification and sea ice melt/formation. The authors compare stations across the Arctic, from the strongly stratified Canada Basin to the weakly stratified Nansen basin. This is a novel and important study that sheds light into how stratification and sea ice melt are closely coupled. There were however some gaps in the research and some confusing wording, which I will explain below. I think this manuscript can be published in Ocean Science after moderate revisions.**

Thanks for the comments, which encourage us to improve the quality of the MS. In the following, we provide point to point responses (blue text) to the suggestions and revised the MS *(black bold italic)* accordingly.

**Major comments:**

M**y first major comment** is that important details were left out of section 2. Specifically, I was left wondering the following:

1.How long were the simulations run for?

We are sorry for missing those important details. All the simulations run for a year, starting on 1 May. We revised the sentence in **section 2.4.**

**Lines 159-160: … *a total of six experiments were conducted at each station for a simulation period of 1 year, starting on 1 May and ending on 30 April next year.***

2.Why were only data for short periods in April or May used for initialization?

The experiments began on May 1st because April or May falls within the late stages of the freezing season or early stages of the melting season in the Arctic Ocean. As a result, the simulation covers a full melting-freezing cycle, which helps to better investigate the feedback effects of melt water on sea ice during summer, as well as its impact on subsequent freezing. This is the reason why we use ITP data from either April or May as initialization for the model. We added sentences in **section 2.4.**

**Lines 162-164: *The experiment started on 1 May with the objective of conducting a full melting period followed by a complete freezing phase in the model, which helps to better investigate the feedback effects of the melt water on sea ice melting in summer, as well as its impact on subsequent freezing in winter.***

3.What time step was used for the simulations?

The timestep is 600s. We revised the sentence in **section 2.4.**

**Line 165: … *the melt water flux of a timestep (600s) is determined by the …***

4.Why were no data used for initialization between 2014 and 2020?

We re-screened the ITPs from 2011 to 2020 and found several usable data from 2015 to 2020 for the experiments. We conducted the experiments again using these data. Due to the addition of some new experiments, we reorganized the station names in the Arctic basin. The stations' location and time information of the ITPs are plotted in Figure 1 and listed in Table 1 in the revised MS.

[Figure]

*Figure 1: Locations of the ITP data used as initial profiles in the model. Stations A1-A7 are located in the Amerasian Basin (indicated by the blue dots), and E1-E7 are in the Eurasian Basin (indicated by the red dots). The black dots represent the ITPs used for comparison with the simulations. The green line represents the trajectory of ITP41. The bathymetry is from ETOPO-2. The same atmospheric forcing field, derived from the 2011-2020 average for the specific region outlined by the solid magenta line, is utilized in all experiments.*

**Table 1. Details of the ITP records used in the model**

| Station | ITP number | Time | Comparison ITP number/Time |
|---------|-----------|------|----------------------------|
| A1 | ITP-53 | 2012.5.1-5.5 | |
| A2 | ITP-18 | 2008.4.6-4.7 | |
| A3 | ITP-108 | 2018.5.7-5.13 | ITP-108/2018.1.1-1.31 |
| A4 | ITP-41 | 2011.5.1-5.15 | ITP-41/2011.8.1-8.5 |
| A5 | ITP-105 | 2019.5.2-5.10 | |
| A6 | ITP-48 | 2012.5.9-5.23 | ITP-48/2012.1.18-1.31 |
| A7 | ITP-47 | 2011.4.12-4.30 | |
| E1 | ITP-93 | 2016.5.1-5.8 | |
| E2 | ITP-57 | 2013.5.25-5.28 | ITP-58/2013.3.9-3.10 |
| E3 | ITP-83 | 2015.5.25-5.30 | |
| E4 | ITP-74 | 2014.5.1-5.2 | ITP-57/2013.8.1-8.2 |
| E5 | ITP-58 | 2013.5.1-5.2 | |
| E6 | ITP-74 | 2014.5.26-5.30 | ITP-74/2014.8.1-8.5 |
| E7 | ITP-111 | 2020.5.25-5.30 | |

5. Why was the MLD definition selected? See Peralta-Feriz and Woodgate (2015; Seasonal and interannual variability of pan-Arctic surface mixed layer properties from 1979 to 2012 from hydrographic data, and the dominance of stratification for multiyear mixed layer depth shoaling - ScienceDirect) for an overview of different MLD definitions in the Arctic.

Thanks for the comments. We chose the threshold criterion Δσ =0.03 kg/m³ for the definition of the MLD with reference to Peralta-Feriz and Woodgate (2015), however, we did not clarify it in the first MS. The following is the reason why we chose the criterion Δσ =0.03 kg/m³ rather than Δσ =0.1 kg/m³.

Peralta-Feriz and Woodgate (2015) chose the threshold criterion Δσ =0.1 kg/m³ because it provides a better comparison with the heuristic method in the pan-Arctic profiles. However, in small parts of the Eurasian Basin where the stratification is very weak, the threshold criterion of Δσ = 0.1 kg/m³ yields MLD estimates more than 20 m deeper than those from a heuristic assessment of the profiles. Thus, for these profiles, they chose a threshold criterion of Δσ = 0.03 kg/m³ instead, which gives a closer agreement with the heuristic assessment.

Based on above information, we compared the two threshold criterions of Δσ = 0.1 kg/m³ and 0.03 kg/m³, and found that Δσ = 0.03 kg/m³ is more consistent with our model results. In the Canadian Basin, there is no obvious difference between the two criterions (fig. 1g-i and fig. 2a-c). However, for stations in the Eurasian Basin, the criterion of Δσ =0.1 kg/m³ overestimates the MLDs. It is found from the winter vertical temperature and salinity profiles that the Atlantic water in the MWP-0% and MWP-20% runs at stations E6 and E7 mixes upwards in winter (Fig.1e and f, below), and the mixed layer reaches 300 m. The MLDs we calculated using the criterion of Δσ = 0.03 kg/m³ is consistent with the modeled temperature and salinity profile (Fig. 2e and f below).

Based on the criterion of Δσ = 0.1 kg/m³, the calculated MLDs for the MWP-40% runs could reach 300 m in March (Fig. 3e and f, below). However, the model results show that the temperature and salinity profiles of the MWP-40% runs in March only mixed to the depths of no more than 200 m (Fig. 1e-f and k-l, below).

[Figure]

**Figure 1: Simulated temperature (top row) and salinity (bottom row) profiles of MWP runs and control runs in mid-April for stations A2, A4, A6, E2, E6 and E7.**

[Figure]

**Figure 2: Time series of the MLD (criterion Δσ =0.03 kg/m³) of the control and MWP runs for stations A2, A4, A6, E2, E6 and E7. The color of each line represents the MWP run factor.**

**Figure 3: Same as figure 1 but for criterion Δσ =0.1 kg/m³.**

In order to clarify the reasons for the choice of the threshold criterion Δσ = 0.03 kg/m³, we added some sentences in **section 2.4.**

**Lines 180-188:** *In this paper, we define the base of the mixed layer (ML) as the depth at which the potential density relative to 0 dbar initially surpasses the shallowest sampled density by the threshold criterion of Δσ =0.03 kg m⁻³. Peralta-Ferriz and Woodgate (2015) used a threshold criterion of Δσ =0.1 kg m⁻³ to define the ML and report the ML properties of the Arctic Ocean from 1979-2012. They also acknowledged that the Δσ =0.1 kg m⁻³ criterion would overestimate the ML in parts of the Eurasian basin where the upper ocean's stratification is very weak. To account for this, they used a threshold of Δσ =0.03 kg m⁻³ in areas with weak upper ocean stratification. We calculated the simulated MLDs using the two methods, respectively, and found that in the Canadian Basin, the MLDs determined by the two criteria show no significant difference. However, in the Eurasian basin, the criterion of Δσ = 0.1 kg m⁻³ would severely overestimate the MLDs. Therefore, we chose the criterion of Δσ = 0.03 kg m⁻³ to determine the MLD in this study.*

My **second major comment** is that there was no discussion about how heat released from the NSTM melts ice through winter. A number of studies (Winter sea-ice melt in the Canada Basin, Arctic Ocean - Jackson - 2012 - Geophysical Research Letters - Wiley Online Library; The impact of stored solar heat on Arctic sea ice growth - Timmermans - 2015 - Geophysical Research Letters - Wiley Online Library; Episodic Reversal of Autumn Ice Advance Caused by Release of Ocean Heat in the Beaufort Sea - Smith - 2018 - Journal of Geophysical Research: Oceans - Wiley Online Library) have shown that strong stratification can store the NSTM through winter. This heat can be gradually released throughout

winter via storm-driven mixing, which breaks down the stratification. Evidence of the heat release can be seen by ocean to ice heat fluxes.

Thanks for the suggestion. We added discussions about how heat released from the NSTM melts ice in winter in the Introduction with reference to the listed previous studies. We added some sentences in **section 1 and section 3.2.2 (b).**

**Section 1, lines 43-46:** *In winter, surface fresh water is recycled via ice formation, weakening ocean stratification (Peralta-Ferriz and Woodgate, 2015), meanwhile, vertical convection caused by brine rejection or storm-driven mixing, can erode the NSTM layer, entraining warm water upward, and impeding winter ice formation (Steele et al. 2011; Jackson et al. 2012; Timmermans, 2015; Smith et al., 2018).*

**Section 3.2.2 (b), lines 364-367:** *In particular, the reduction in summer melt water leads to a weakening or even absent NSTM, and that there isn't enough heat stored in the subsurface layer to replenish the heat loss at the surface when autumn arrives, leading to a more rapid cooling of water temperature to the freezing point in autumn, and hence increasing ice formation in autumn. This result suggests that the presence of the NSTM effectively hinders sea ice growth in autumn, which is consistent with Toole et al. (2010)'s results.*

I wonder if including ITP data from the winter of 2007-2008 – a year where the NSTM was stored year-round – would change the results in Figure 5c?

Thanks for the suggestion. We studied the ITP data (ITP18) located in the Canadian Basin from 2007 to 2008 and found that a year-round NSTM. Figure 4 (below) shows the vertical temperature-salinity profile of ITP18 in April 2008. There were two temperature maxima in the upper 100 meters: one at around -80 meters, which is the Pacific Warm Water, and the other at around -40 meters, which is NSTM. We conducted an experiment in the Canadian basin (A2 in the new MS), initializing the model using data from April 2008 (ITP-18).

[Figure]

**Figure 4: Initial profile of station A2 in the new MS.**

The experiments using ITP18 as the initial field did not change the results of Figure 5c (Figure 6c in the new MS). We added this experiment to the new MS and added sentences to describe ITP18 (station A2 in the new MS) in **Section 2.2**.

**Lines 130-133:** *The temperature profile at station A2 shows two peaks in the upper layer, one is the NSTM, and the other is the PSW. The initial profile at A2 station was obtained from ITP measurements in the southern Canadian Basin in 2007-2008, and due to a strong halocline that year, the NSTM formed in the summer of 2007 persisted until the spring of 2008 (Jackson et al., 2012).*

In fact, we originally intended to conduct the experiment using the data from April 2007, because we wanted to know if the NSTM stored until winter 2008 would change the results of Figure 5c (Figure 6c in the new MS). However, the data of ITP18 starts in August 2007. So, we conducted another experiment using the profile of ITP18 at the end of summer 2007 as the initial field. The experiment results show that the NSTM could be preserved until winter 2008, but its ocean-ice heat flux did not show episodic high values (Fig. 5 below). This is because the episodic high winter ocean-ice heat fluxes are usually associated with strong wind events (Smith et al., 2018). In this study, all experiments utilized regionally averaged wind fields to eliminate the impact of wind field variability. This may be the reason why this one-dimensional model did not simulate episodic high values of the ocean-ice heat flux in winter 2008.

[Figure]

**Figure 5: Experimental results using ITP18 data from mid-September 2007 as the initial field. (a) effective sea ice thickness; (b) ocean-ice heat flux; (c) temperature time series in the upper 100 m.**

We added some sentences in **section 3.1.2** to explain why our model does not reproduce the episodic high values in winter.

**Lines 253-261:** *As observed by Jackson et al. (2010) and Steele et al. (2011), our model also shows that the NSTM normally deepens, cools, and disappears throughout the autumn and winter (Fig. 4). However, it has been observed as a year-round feature sometimes. Jackson et al. (2012) found that when ITP18 drifted into shallow waters from early to mid-December, the ocean-ice heat flux reached up to 55 W m$^{-2}$ (Jackson et al., 2012), reduced sea ice thickness at the end of the 2008 growth season by about 25% (Timmermans, 2015). Smith et al. (2018) also discovered occasional high values of the winter ocean-ice heat flux (about 100 W/m$^2$) in the Canadian Basin using ITP and CTD data from 2015. These high winter sea-ice heat fluxes are usually associated with strong wind events (Smith et al., 2018). In this study, all experiments utilized regionally averaged wind fields to eliminate the impact of wind field variability. This may be the reason why our one-dimensional model did not reproduce the episodic high values of the ocean-ice heat flux in winter successfully.*

My **third major concern** confusion with the feedback language used - in general, I got confused with how the authors were referring to a negative and positive feedback in this context. I think adding some clear definition of negative and positive feedbacks in the introduction as well as how these feedback are related to the atmosphere-ocean-ice- system would help the readers understand the importance of these results and how they fit into previous knowledge.

Thanks for this suggestion. We are sorry for missing some important references. We revised sentences in **section 1.**

**Lines 47-57:** *Melt water from the sea ice has a comparatively low density and therefore accumulates in the top ocean layer, strengthens the upper ocean stratification. Due to the stabilizing of the cold halocline, the ocean heat flux available to melt sea ice decreases, which in turn hinders sea ice melting. This is a negative sea ice/ocean feedback on sea ice melting (Bintanja et al., 2013), and we call it melt water feedback in this paper. Zhang (2007) and Bintanja et al. (2013) suggest that this negative sea ice/ocean feedback can explain the anomalous increase in Antarctic sea ice extent before 2010s. Many positive feedback processes in the Arctic atmosphere-ocean-ice systems are extensively studied, such as the well-known sea ice albedo feedback (Hall, 2004; Winton, 2000; Pithan and Mauritsen, 2014), water vapor feedback (Gordon et al., 2013; Taylor et al., 2013) and the Cloud-Albedo feedbacks (Zelinka et al., 2012; Bodas-Salcedo et al., 2016). However, there are almost no quantitative studies on this negative melt water feedback on sea ice melting in the Arctic, although many previous studies have investigated the effects of increased freshwater flux by adding freshwater flux to the ocean surface in models to represent increased runoff or precipitation (Nummelin et al., 2015, 2016; Davis et al., 2016a; Pemberton and Nilsson, 2016).*

Furthermore, the effect of melt water on ice freezing in winter cannot be considered as a feedback process and we revised the relevant content in the new MS.

**Abstract, lines 12-15:** *The impact of melt water released during the previous melting season on ice growth in winter depends on the strength of stratification. After removing all the melt water during the summer, ice formation in areas with strong stratification increased by 12.3% during the winter, while it decreased by more than 40% in areas with weak stratification.*

S**ection 3.2.2 (b), lines 374-375:** *The results indicate that the impact of melt water released during the previous melting season on winter sea ice growth depends on the strength of stratification, with gradually transitions from promoting to impeding ice growth as the halocline weakens.*

S**ection 3.2.2 (c), lines 385-386:** *The impact of melt water released during the previous melting season on the subsequent winter ice formation depends on the strength of stratification. It hinders (promotes) ice formation in areas with strong (weak) stratification.*

Section 5, lines 487-489: *Our findings reveal that the effects of melt water from the previous melting season on the subsequent winter ice formation depend on the strength of stratification. Specifically, it impedes ice formation in areas with strong stratification, while it promotes it in areas with weak stratification.*

My **fourth major concern** is that knowledge from several important manuscripts were missing. In addition to the above cited manuscripts, I suggest the authors read the following:

(1) Toward Quantifying the Increasing Role of Oceanic Heat in Sea Ice Loss in the New Arctic in: Bulletin of the American Meteorological Society Volume 96 Issue 12 (2015) (ametsoc.org)

(2) Modeling the formation and fate of the near-surface temperature maximum in the Canadian Basin of the Arctic Ocean - Steele - 2011 - Journal of Geophysical Research: Oceans - Wiley Online Library

(3) Winter Convection Transports Atlantic Water Heat to the Surface Layer in the Eastern Arctic Ocean in: Journal of Physical Oceanography Volume 43 Issue 10 (2013) (ametsoc.org)

(4) Freshwater and its role in the Arctic Marine System: Sources, disposition, storage, export, and physical and biogeochemical consequences in the Arctic and global oceans - Carmack - 2016 - Journal of Geophysical Research: Biogeosciences - Wiley Online Library

(5) Arctic sea-ice melt in 2008 and the role of solar heating | Annals of Glaciology | Cambridge Core
We are sorry for missing these important references. We read and cited the references in **sections 1.**

**Lines 29-30:** *Ocean-ice heat fluxes play a crucial role in modulating the Arctic Sea ice growth/melt cycle, with half of the total heat flux absorbed by the sea ice originating from the ocean (Carmack et al., 2015).*

**Line 46:** *… entraining warm water upward, and impeding winter ice formation (Steele et al. 2011; Jackson et al. 2012; Timmermans, 2015; Smith et al., 2018).*

**Lines 70-72:** *Previous research suggests that brine-driven surface convection could entrain the Atlantic Water heat upwards in the Eurasian Basin (Polyakov et al., 2013a, 2020), while the strong stratification impede this convection process in the Canada Basin (Toole et al., 2010).*

**Line 28:** *… and seasonal ice melt are critical factors that maintain this stratification (Haine et al., 2015; Carmack et al., 2016).*

**Sections 3.1.2, line 253:** *… the main heat source is the absorption of solar radiation (Perovich et al., 2011) …*

**Minor comments**

- Section 2.1 – I find this section confusing the way it is written. Specifically, the variables were not described in the order that they appeared so I had to jump around to understand. I suggest reorganizing so that the variables are described. Another option is to add a table that lists and defines all variables.

We are sorry for the confusion in the description of variables, and we reorganized the variable description in **section 2.1, lines 95-109:**

*…. The heat fluxes at the ice top and bottom are:*

$$F_{top} = F_s(\alpha) - Fs_{ice} \tag{1}$$

$$F_{bot} = Fb_{ice} - F_b \tag{2}$$

*where $F_s$ is the surface heat flux absorbed by the ice, $Fs_{ice}$ is the conductive heat flux from the upper layer of the sea ice to the ice surface, $Fb_{ice}$ is the conductive heat flux from the ice bottom to the lower layer of the sea ice. $F_b$ is the ocean-ice heat flux:*

$$F_b = c_{sw}\rho_{sw}\gamma(T_{sst} - T_f)u^* \qquad\qquad (3)$$

*where $\gamma$ is the heat transfer coefficient and $u^*$ is the frictional velocity between ice and water.*

*The albedo parameterization of this model is dependent on ice thickness:*

$$\alpha = \alpha_{i_{min}} + (\alpha_{i_{max}} - \alpha_{i_{min}})(1 - e^{-h_i/h_\alpha}) \qquad\qquad (4)$$

*where $\alpha_{i_{min}}$=0.08 and $\alpha_{i_{max}}$=0.64 are the maximum and minimum ice albedo values, respectively $h_\alpha$=0.65 is the ice thickness for albedo transition, and $h_i$ is the ice thickness.*

*The net ocean surface heat flux can be written simply as Steele et al., (2010):*

$$F_{ocean} = F_{sw} + F_b + F_{ao} \qquad\qquad (5)$$

*where $F_{sw}$ is the heat flux from solar radiation, $F_b$ is the ocean to ice heat flux, and $F_{ao}$ is the heat flux from the ocean to the atmosphere through the ice-free area (including longwave radiation and sensible and latent heat flux).*

- Line 74 – What are the estimates of vertical diffusivity in the Canada Basin versus the Nansen Basin? I imagine vertical diffusivity is smaller in the Canada Basin due to the strong stratification. I suggest the authors add a sentence that cites studies that have examined vertical diffusivity from different regions to justify their decision to choose this parameter value.

  Thanks for the comments. We cited previous studies and revised the sentence in **section 2.1.**

  **Lines 85-90:** *Shaw and Stanton (2014) show that the vertical diffusivity in the deep central Canadian Basin averages near-molecular levels, ranging between $2.2\times10^{-7}$ $m^2$ $s^{-1}$ and $3.4\times10^{-7}$ $m^2$ $s^{-1}$, and Fer (2009) found that vertical diffusivity between $10^{-6}$ -$10^{-5}m^2$ $s^{-1}$ in the Eurasian Basin. The background vertical diffusivity of the model used in this study is set to $10^{-6}$ $m^2$ $s^{-1}$, which is a representative value in the central Arctic Ocean and has been applied to several one-dimensional models studying the Arctic Ocean (Linders and Björk, 2013; Nummelin et al., 2015; Davis et al., 2016)*

- Line 81 – Why were these values chosen for albedo. Pleas add some references to justify this choice.

  The choice of albedo parameter in this paper was determined by artificial adjustment. Initially, we used the default value in the model ($\alpha_{i_{min}}$=0.2, $\alpha_{i_{max}}$=0.65, $h_\alpha$=0.5) (Hansen et al., 1983), but this albedo value is relatively high, resulting in little sea ice melt in summer and less shortwave radiation being absorbed by the ocean, making it difficult to simulate the NSTM well. Therefore, after adjustment, we chose these albedo parameters. We cited the reference in **sections 2.1.**

  **Line 106:** …*The albedo parameterization of this model is dependent on ice thickness (Hansen et al., 1983) …*

- Figure 1 – I can't see A5 in this figure.

  Due to the inappropriate color scheme, it is difficult to distinguish A5 from the others in Figure 1 of the the old MS. We redraw Figure 1 in the new MS (see Figure .1 above).

- Line 99 – Why is ITP E6 not listed here?

  Sorry for missing it, we corrected it in **section 2.2.**

  **Lines 113-114:** … *A1-A7 located in the Amerasian Basin (the blue dots in Fig. 1) and E1-E7 in the Eurasian Basin (the red dots in Fig. 1).*

- Lines 112 to 114 – The NSTM is defined based on salinity, and it looks like the temperature maximum in A1 to A3 shown in Figure 2 include both NSTM and Pacific Summer Water. I suggest adding this to the Figure 2 description.

  The NSTM was defined by Jackson et al., (2010) as the shallowest temperature maximum (Tmax)

that satisfies three criteria. These are:

Criterion 1. The Tmax's temperature above freezing must be greater than 0.2°C.

Criterion 2. The Tmax's temperature above freezing must be more than 0.1°C warmer than an immediate underlying temperature minimum.

Criterion 3. The Tmax's salinity must be less than 31.

Based on criterion 2, The temperature profiles in A1, A2 and A3 in the old MS (which are A1, A3, and A4 in the new MS) not include NSTM.

■ Lines 178 to 179 – I think the authors have made an error here. They state that the MLD was deeper in the western Arctic than Eastern Arctic. I think here they define the western Arctic as the Canada Basin? Please clarify.

Thanks for pointing out the error. We revised the sentence in **section 3.1.1.**

**Lines 223-224: *Both the modelling and the observations show that the MLDs are usually deeper in the Eurasian Basin than in the Canadian Basin.***

■ Section 3.1.1 – While the model does a good job of reproducing the NSTM, the modeled NSTM is colder than the observed one, which could have implications for sea ice melt and formation. Could the authors please add a few words about how the colder NSTM could impact their results?

Thanks for this suggestion. We added two sentences in **section 3.1.1.**

**Lines 217-218: *However, the simulated summer NSTM in the Canadian Basin is generally cooler than the observations (Fig. 5a). This discrepancy may lead to an overestimation of winter ice formation in the simulations.***

■ Lines 196 to 203 – Previous literature (e.g., Figure 5b in Jackson et al., 2012; Figure 7 in Smith et al., 2018) show observational values of ocean to ice heat flux, including episodic high values in the Canada Basin in winter. I suggest the authors compare their results with these studies.

Thanks for this suggestion. We added some sentences in **section 3.1.2.**

**Lines 253-261: *As observed by Jackson et al. (2010) and Steele et al. (2011), our model also shows that the NSTM normally deepens, cools, and disappears throughout the autumn and winter (Fig. 4). However, it has been observed as a year-round feature sometimes. Jackson et al. (2012) found that when ITP18 drifted into shallow waters from early to mid-December, the ocean-ice heat flux reached up to 55 W m$^{-2}$ (Jackson et al., 2012), reduced sea ice thickness at the end of the 2008 growth season by about 25% (Timmermans, 2015). Smith et al. (2018) also discovered occasional high values of the winter ocean-ice heat flux (about 100 W/m$^2$) in the Canadian Basin using ITP and CTD data from 2015. These high winter sea-ice heat fluxes are usually associated with strong wind events (Smith et al., 2018). In this study, all experiments utilized regionally averaged wind fields to eliminate the impact of wind field variability. This may be the reason why our one-dimensional model did not reproduce the episodic high values of the ocean-ice heat flux in winter successfully.***

■ Figure 6 – Why is there only 1 station from the Canada Basin?

In the first MS, the simulation results for stations in the Canadian Basin are similar, so we only show one station. In the new MS, we added several experiments and showed the results for three stations in the Canadian Basin (A2, A4 and A6) and three stations in the Eurasian Basin (E2, E6 and E7). We added some sentences in **Section 3.2.1 (a).**

**Lines 272-275: *The experimental results for some stations in this study are very similar, so this paper shows the simulation results for six representative stations to show the general behavior of the***

*model and the impact of ocean stratification. Three of them are located in the Amerasian Basin (A2, A4 and A6) and three in the Eurasian Basin (E2, E6 and E7).*

■ Lines 234 to 240 – How do these results compare with results from Peralta-Feriz and Woodgate, 2015?

The simulated MLDs in the control experiments are in good agreement with the observation-based statistics of Peralta-Feriz and Woodgate (2015). We revised sentences in **section 3.1.1.**

**Lines 219-222:** *In all control runs, the simulated maximum winter MLD is ~33 m in the Canadian Basin, ~43 m in the Makarov Basin, ~67 m in the Amundsen Basin, and more than 100m in the Nansen Basin. These results are comparable to the observations. The observed maximum winter MLDs in Canada and the Makarov Basin are 29 ± 12 m and 52 ± 14 m, respectively, and those in the Eurasian Basin range from ~50 to over 100 m (Shimada et al., 2001; Peralta-Ferriz and Woodgate, 2015).*

■ Figure 11 – It is difficult to distinguish the line colours. Also, as mentioned above, I find the wording of positive and negative feedback to be confusing.

This figure could cause some misunderstandings and does not help readers understand the article. We removed it in the new manuscript, and we revised the feedback language used in this article as mentioned above.

■ Figures 12 and 13 – I found it confusing to decipher what positive and negative values mean. It would help the reader if you added this to the figure caption.

We added explanations for positive and negative values in Figure 12 and 13 in the new MS.

[Figure]

***Figure 12: Time series of (left) the anomalies of ice bottom change rate and (right) the anomalies of ice surface change rate for stations A2, A4, A6, E2, E6 and E7. The anomalies are obtained from the MWP run minus the control run. The negative (positive) values indicate faster (slower) rates of ice decrease in the MWP run compared to the control run. The color of each line represents the MWP run factor.***

[Figure]

*Figure 13: Time series of (left) the anomalies of ocean-ice heat flux, (middle) the anomalies of shortwave radiation, and (right) the anomalies of ocean-atmosphere heat flux for stations A2, A4, A6, E2, E6 and E7. The anomalies are obtained from the MWP run minus the control run. The negative (positive) value indicates heat gain (loss) by the ocean in the MWP run compared to the control run. The color of each line represents the MWP run factor.*

■ One suggestion I have for future research, which could be added as a sentence in the discussion, is to add wind sensitivity experiments to explain how wind mixing impacts the stratification and sea ice melt/formation.

We added two sentences in **section 4.**

**Lines 453-457:** *In addition, as mentioned in section 3.1.2, changes in wind speed will affect stratification and the melt/formation of sea ice through increased vertical mixing. The ideal modelling method used in this study cannot reproduce the episodic high values of ocean-ice heat flux caused by wind mixing, as reported by Jackson et al., (2012) and Smith et al., (2018). Therefore, it is necessary to consider the wind speed on the role of melt water in the sea ice-ocean coupled system in future work.*

---

## Author Comment (AC3)

This is my first review of a paper titled 'A numerical study on melt water feedback in the coupled Arctic Sea ice-ocean system' by Zhang et al. The paper discusses the impact of freshwater from sea ice melt on sea ice growth/melting itself. Using a 1D sea ice - ocean model the authors find that depending on the initial stratification, the melt water has a strong negative feedback on ice melt but depending on stratification can have either a positive or negative impact on the following winter ice growth. The results themselves are interesting, although I do question the linkage to meltwater alone (see below). As such the manuscript lacks some details and could be published with minor revisions, although the results reflect the importance of freshwater in general, not melt water alone. However, I give suggestions to modify their experimental setup to actually attribute their results to sea ice meltwater, or to carry out with their results, but to recast their results in terms of general freshwater perturbations. I encourage the authors to consider the suggested modification to their approach and therefore suggest major revision.

Thank you very much for the comments and suggestions, which help us to improve the quality of the MS. In the following, we provide responses (blue text) to the comments, and revised the MS accordingly *(black bold italic)*.

**Major comments:**

**1)** If I understand correctly, what the authors do here, is that they have one 'control' simulation with full meltwater release and perturbation experiments with scaled down versions of meltwater release. However, although the amount of meltwater release is scaled down the sea ice melt itself stays the same, i.e. some freshwater disappears in the process.

Because the authors also don't take into account any other sources of freshwater, the influence of sea ice meltwater in respect to other freshwater sources remains unclear. To me, the experiments, as they are done now, appear more as traditional freshwater release experiments (albeit with seasonal cycle), rather than experiments that would try to isolate the role of sea ice meltwater. If the authors would want to really isolate the role of sea ice meltwater on the ice melt/growth, then I think one would need to do something like this:

a) Create a simulation that reproduces ITP profiles with other freshwater fluxes included. Based on Figure 4 the authors claim that this is the case already, but the simulations done here are very short and already at the surface mixed layer the salinity differences can be several PSU in some stations. The easiest would be to deduce freshwater convergence from ITPs and remove the (observed) sea ice melt water flux from the convergence, allowing the model to calculate that. It would also be interesting to diagnose the actual sea ice melt water flux from ITPs and for example cross-correlate that with sea ice growth across the different ITP's to see if the authors hypothesis can be identified in the observations.

b) Once the authors have the stable control simulation, they could repeat the experiments, keeping the freshwater convergence the same, but perturbing the sea ice meltwater flux.

Such an experimental setup would answer the question 'what is the importance of sea ice meltwater for sea ice melt/growth?'. In the current setup I would argue that all the authors can truly answer to is 'what is the role of freshwater for sea ice melt/growth (thermodynamic)?'. I do think the identified feedbacks are neat, but in the current form I think the authors would need to rephrase their aim and discuss the caveats of their experimental setup. For example, in their 0% experiment, there is no freshwater source to the surface, which obviously gives a very large signal, but it is not realistic to claim that this signal can be attributed to sea ice meltwater (because there would be other freshwater

sources contributing to the stratification). I leave it to the authors to decide on their approach, but I would think modifying their setup would be achievable and would certainly increase the impact of the paper.

Thanks for the comments and experiment suggestions. The aim of this study is to investigate the melt water feedback on sea ice melting, so we remove sea ice melt water in the model directly to disable the feedback mechanism to examine melt water feedback effects by comparing it with the control run. In order to avoid the exaggeration of the meltwater feedback effect, we added other external freshwater forcing in all the experiments in the new MS. We tried to deduce freshwater convergence from an ITP and found it difficult to obtain a real sea ice melt water flux. The suggested experiment design is good to study the role of the sea ice melt water in the sea ice melting/growth but may not be suitable to study the meltwater feedback. Perhaps we did not claim the aim of our study clearly, we revised the MS and clarified our research objective in the **Section 1.**

**Lines 47-57:** *Melt water from the sea ice has a comparatively low density and therefore accumulates in the top ocean layer, strengthens the upper ocean stratification. Due to the stabilizing of the cold halocline, the ocean heat flux available to melt sea ice decreases, which in turn hinders sea ice melting. This is a negative sea ice/ocean feedback on sea ice melting (Bintanja et al., 2013), and we call it melt water feedback in this paper. Zhang (2007) and Bintanja et al. (2013) suggest that this negative sea ice/ocean feedback can explain the anomalous increase in Antarctic sea ice extent before 2010s. Many positive feedback processes in the Arctic atmosphere-ocean-ice systems are extensively studied, such as the well-known sea ice albedo feedback (Hall, 2004; Winton, 2000; Pithan and Mauritsen, 2014), water vapor feedback (Gordon et al., 2013; Taylor et al., 2013) and the Cloud-Albedo feedbacks (Zelinka et al., 2012; Bodas-Salcedo et al., 2016). However, there are almost no quantitative studies on this negative melt water feedback on sea ice melting in the Arctic, although many previous studies have investigated the effects of increased freshwater flux by adding freshwater flux to the ocean surface in models to represent increased runoff or precipitation (Nummelin et al., 2015, 2016; Davis et al., 2016a; Pemberton and Nilsson, 2016).*

We originally intended to conduct experiments using the suggested experimental design. We calculated the freshwater convergence of the ITPs (Fig.1 below). However, the actual amount of ice melt corresponding to each ITP data point is unknown, which makes it difficult to accurately calculate the proportion of melt water in the freshwater convergence. Furthermore, ITPs is continuously moving. Advection and other external forcing processes have a significant impact on the vertical temperature and salinity, which leads to large fluctuations in the calculated freshwater flux value (black solid line in Fig.1 below). In the coupled ice-ocean model, the calculation of melt water flux for each time step depends on the sea ice melting, and the removal of a portion of freshwater flux based on the freshwater convergence calculated from ITP will lead to an inaccurate assessment of the melt water feedback in the coupled model.

[Figure]

**Figure 1. The time series of freshwater flux calculated for station A4 and ITP41. The black solid line represents the unsmoothed freshwater flux values of ITP41, while the blue dotted line shows the 30-day moving average of ITP41. The blue solid line represents the simulated freshwater flux values at station A4.**

We acknowledge that the signal of the melt water feedback is to be exaggerated if the other external freshwater forcings were ignored. So, we re-run all experiments with incorporating other fresh water sources. According to Haline et al. (2015)'s research on the freshwater balance of the Arctic Ocean, the main sources of freshwater input in the Arctic Ocean are river runoff, Bering Strait inflow, and net precipitation, with a total input of approximately 9400±490 $km^3yr^{-1}$. Meanwhile, freshwater output occurs through the Davis Strait, the Fram Strait, and the Canadian Archipelago, with a total output of approximately 8200±550 $km^3yr^{-1}$. The net input of freshwater into the Arctic Ocean is approximately 1200 $km^3yr^{-1}$. We added this part of the freshwater as other freshwater sources into the model.

With the addition of the external freshwater input, the effect of melt water feedback decreased slightly compared to the old experiments without other external freshwater sources. However, it has little impact on the conclusions. We added some sentences in **section 1** and **section 2.3** to illustrate the existence of external freshwater forcing in the model.

**Section 1, lines 38-43:** *The volume of net input freshwater that comes to the Arctic Ocean from external sources is roughly 1200±730 $km^3$ annually, with an inflow of 9400 ± 490 $km^3yr^{-1}$ and an outflow of 8250 ± 550 $km^3yr^{-1}$, while approximately 11300 $km^3$ freshwater enter the ocean in summer through melting. The volume of melt water in the Arctic Ocean is approximately 10 times greater than that of net freshwater input, leading to an increase of 1.2 m in the Arctic Ocean's surface freshwater layer (Haine et al., 2015), which separate the surface ML from the near-surface temperature maximum (NSTM).*

**Section 2.3, lines 152-157:** *Although the focus of this study is on the melt water feedback in the coupled ocean-sea ice system, freshwater fluxes due to runoff inflow, precipitation minus evaporation, and input or output from straits also contribute to the stratification changes of the Arctic Ocean. The signal of the melt water feedback is to be exaggerated if those external freshwater forcings were ignored. So, we also consider the external freshwater forcing in the experiments. Haine et al. (2015) reported that the annual net inflow of freshwater to the Arctic Ocean is approximately 1200 $km^3yr^{-1}$, and we add this net freshwater inflow to our model on a daily average to represent various freshwater sources other than the melt water.*

We revised the values of the experimental results in **Abstract.**

**Line 12:** *… by 16.6% by strengthening ocean stratification.*

**Lines 14-15:** *… by 12.3% during the winter, while it decreased by more than 40% in areas with weak stratification.*

We also made changes to other similar values in the new MS, but since these changes are scattered throughout in the MS, we are not list them here.

We would like to clarify that this study is based on ideal one-dimensional model experiments where each experiment uses identical atmospheric and external freshwater forcing fields. So, we do not aim to perfectly replicate the variability of the ITP profiles in the control experiments. The comparison between the observations and simulations shown in Figure 4 of the first MS (but it is Figure 5 in the new MS) is to demonstrate that the seasonal changes of ocean temperature and salinity simulated by our one-dimensional model are qualitatively consistent with the observed data. This is to verify that our experimental results are not deviating from reality. Such a model validation method has also been used in some previous studies of Arctic Ocean stratification using 1D models (such as Linders and Björk, 2013; Toole et al., 2010). We added a figure (below, fig. 2, but it is figure. 4 in the new MS) in the new MS to compare the time series of simulated and observed temperature and salinity. This figure aims to illustrate the temperature and salinity changes simulated by the model in detail and further demonstrate that our model results are reliable.

[Figure]

Figure 2 (Figure 4 in the new MS): The time series of temperature (left) and salinity (right) for the upper 50 meters were derived from (a), (b): ITP41 observations and (c), (d): simulated values at station A4, respectively.

We revised some sentences in **section 3.1.1.**

**lines 197-216:** *Figures 4 and 5 show the comparison between the simulated temperature and salinity profiles of the control runs and the ITP observations (the details of the 6 ITPs datasets for comparison with the simulated results are listed in Table 1). The results of the one-dimensional model reasonably reproduce the seasonal variations of the vertical temperature and salinity*

*structure in the Arctic Ocean. It should be noted that this study does not aim to perfectly replicate the variability of the ITP profiles, as the variability of the Arctic Ocean temperature and salinity structure is influenced not only by surface freshwater fluxes but also by an array of external local forcings, such as high-frequency variations in wind fields, local precipitation or evaporation, horizontal transport of freshwater, and observational errors. Despite some discrepancies between the simulated and observed vertical profiles, the simulations of these ideal experiments are still qualitatively consistent with the observations. Therefore, the simulation results obtained in this study are reliable.*

*ITP41 measured relatively complete temperature and salinity data along its pathway (green line in Fig. 1) in the Canadian Basin from May 2011 to April 2012, and the data measured by ITP41 in May 2011 also serve as the initial field for station A4 in the model. Therefore, we compared the complete time series of the temperature and salinity of the ITP41 observations with the simulations. Both the observations and simulations show that large quantities of freshwater, primarily melt water, cover the ocean surface during the melting season, typically lasting from June to September. As a result, a significant salinity gradient forms between the surface water and underlying water layers, creating a new, fresher surface layer (Fig. 4b and d). And the model also successfully reproduces the NSTM at the base of the summer ML, present at approximately 10-20 m (Fig. 4a and c). During the freezing season (October to the next April), brine rejection enhances the turbulence scale perturbations, leading to a deeper ML, and the NSTM generated during the summer progressively cools and vanishes (Fig. 4a and c).*

*Furthermore, we compared the simulated values with actual summer and winter observations gathered from select stations in the vicinity of the simulation. Figure 5 shows …*

**2)** L74 in the model description the authors write that the sea ice package is based on viscous-plastic sea ice model. Although this is true, perhaps there should be a sentence specifying that in 1D case the dynamics don't play a role (or do they?) and the ice growth is determined by thermodynamics alone.

We added a sentence in **section 2.1.**

**Lines 92-93:** *Although the one-dimensional model includes a dynamics sea ice module, sea ice changes are only determined by thermodynamics processes.*

**3)** Ice thickness initialization to 2.5 m is an idealization (of multiyear ice), and that is fine as such, but I'd imagine the simulations are relatively cheap to do so I wonder if it would be worth repeating the experiments with thinner initial ice (something that represents first year ice). I would think that most locations in the Eurasian basin rarely have 2.5 m thick ice these days.

This suggestion is very helpful. We added experiments with thinner ice (1.5m) at four stations A3, A6, E2 and E7. We found that the feedback of melt water on sea ice melting is not significant in summer. however, it is more effective in hampering upward mixing of Atlantic water and melting sea ice during winter. We added **section 3.3 (lines 390-422)** to discuss the thinner ice experiments.

*3.3 Sensitivity experiments with thinner sea ice*

*In recent decades, it has been observed that Arctic summer sea ice appears to be decreasing rapidly (Perovich et al., 2019), with larger ice-free areas in summer and thinner winter sea ice (Haine and Martin, 2017). Thus, several experiments are conducted using thinner initial ice (1.5 m). To highlight the effects of strong or weak CHL, we selected stations A3, A6, E2 and E7 to do the thinner ice experiments.*

*In the control run, the initial thinner ice of 1.5m completely melts in late July (Fig. 14a), and*

*the maximum ocean-ice heat flux can reach 330Wm⁻² (Fig. 14b). During winter, E7 station produces less sea ice because it possesses a weaker stratification (see Fig. 14a), which is consistent with experiments that had an initial ice thickness of 2.5 m.*

*Compared to the control runs and the MWP20%-80% runs, the sea ice melts more slowly in the MWP-0% runs (Figures 14c-f), which contrasts with the experiments with a thicker initial ice. This may be due to the fact that the thinner initial ice contribute to the presence of a larger open ocean during the summer and increased wind input enhances the mixing level, resulting in more heat being mixed into the deeper ocean. As a result, the heat available for melting sea ice is reduced. Figures 15a-d clearly demonstrate the process by late July, the temperature of the upper ocean is remarkably lower in the MWP-0% runs, while the temperature below 10m is considerably higher compared to the other runs.*

*During winter, the role of melt water in hindering the upward mixing of AW is more evident in the thinner initial ice experiments. Removing 40% of melt water during the summer in the thinner initial ice runs can enable the upward mixing of the AW (Fig. 16d and h) and subsequent melting of sea ice in winter (Fig. 14f and Fig. 17b). However, it would require the thicker initial ice runs to remove over 80% of melt water to achieve similar results (Fig. 9f and l).*

*The thinner ice experiments indicate that as multi-year ice in the Arctic Ocean is replaced gradually by seasonal sea ice, melt water will play a more significant role in impeding vertical mixing and winter ice melting in the future.*

[Figure]

*Figure 14: Time series of the (a) effective sea ice thickness and (b) ocean-ice heat flux (negative values represent the heat transfer from ocean to ice) for control runs with thinner initial ice thickness. The subplot in (b) shows the time series of ocean-ice heat fluxes between May and August, indicating that ocean-ice heat fluxes can reach a maximum of 330Wm⁻². (c)-(f): Time series of the anomalies of effective ice thickness for stations A3, A6, E2 and E7. The anomalies are obtained from the MWP run minus the control run.*

[Figure]

*Figure 15: Simulated temperature (top row) and salinity (bottom row) profiles of MWP runs and control runs in late-July for stations A3, A6, E2, and E7 of the thinner initial ice experiments.*

[Figure]

*Figure 16: Same as figure 15 but in mid-April.*

[Figure]

*Figure 17: Same as figure 11 but for stations A3, A6, E2 and E7 of the thinner initial ice experiments.*

We added a sentence in **section 5.**

**Lines 490-492:** *3. Sensitivity experiments with thinner initial ice indicate that as multi-year ice in the Arctic Ocean is gradually replaced by seasonal sea ice, melt water will play a more significant role in hindering vertical mixing and winter ice melt in the future.*

**4)** Similar to the comments by the other reviewer, the model experiment need to be better documented (when are they initialized, how long are they run for etc.). Some of this information is in discussion section, but that comes far too late for the reader.

As requested by Reviewer #1, we supplemented the missing experimental information by adding several sentences in **section 2.4.**

**Lines 159-160:** *To investigate the impact of the release of melt water on ocean stratification and sea ice, a total of six experiments were conducted at each station for a simulation period of 1 year, starting on May 1 and ending on April 30 next year.*

**Lines 162-164:** *The experiment started on 1 May with the objective of conducting a full melting period followed by a complete freezing phase in the model, which helps to better investigate the*

*feedback effects of melt water on sea ice melting in summer, as well as its impact on subsequent*
*freezing in winter.*

**Line 165:** *… the melt water flux of a timestep (600s) is determined by the freshwater content of*
*the …*

**5)** I would change the order of discussions and conclusions.

Thanks for the suggestion. We changed the order of the discussion and conclusion.

**6)** Figure 3: are the labels in f and g correct, or should they be the other way around?

We made a writing error in the caption where we wrote (g) instead of (f). We corrected this error
in the new MS. We apologize to the reviewers for any inconvenience caused by this error.

**7)** Figure 5 and other similar figures: I would encourage the authors to show anomalies from a control
case instead of the full values (it is hard to appreciate the differences at the moment).

We redraw these figures in the new MS to show the anomalies.

[Figure]

*Figure 6: Time series of the (a) effective sea ice thickness (Hice), (b) ice concentration (Aice) and (c) ocean-ice*
*heat flux (Fb, negative values representing the heat transfer from the ocean to the ice) for all control runs. The*
*amplified subplot shows the anomalies (each control run minus the average of all control runs) during the*
*months of February to April.*

[Figure]

*Figure 10: Time series of (left) the anomalies of effective ice thickness and (right) anomalies of ice concentration for stations A2, A4, A6, E2, E6 and E7. The anomalies are obtained by the MWP run minus the control run.*

[Figure]

*Figure 13: Time series of (left) the anomalies of ocean-ice heat flux, (middle) the anomalies of shortwave radiation, and (right) the anomalies of ocean-atmosphere heat flux for stations A2, A4, A6, E2, E6 and E7. The anomalies are obtained from the MWP run minus the control run. The negative (positive) value indicates heat gain (loss) by the ocean in the MWP run compared to the control run. The color of each line represents the MWP run factor.*

---

## Author Response (AR2)

**Responses to Reviewer #1**

Second review of 'A numerical study on melt water feedback in the coupled Arctic Sea ice-ocean system' by Zhang, Bai, and Wang. I appreciate the detailed responses to my comments from the first round of reviews. The authors did a very nice job of addressing my concerns and improving the manuscript.

Thank you for your comments, which have greatly improved the quality of this MS. In the following, we provide point to point responses (blue text) to the suggestions and revised the MS *(black bold italic)* accordingly.

**1.** One area that I still think needs improvement is the way that negative and positive feedbacks is explained. For example, lines 11-12 state '2) Melt water release has negative feedback on ice melting' and I was left struggling to understand what negative feedback means in this scenario. Does it mean that less ice is melting? Or that more ice is melting. I wonder if the authors could remove wording relating to feedback and just simply talk about the processes. For the example given, lines 11-12 could simply state '2) Melt water reduced ice melting by 16.6% by strengthening ocean stratification'.

Sorry for any confusion caused by wording. In this study, the negative feedback means less ice melting. To explain it more directly, we eliminated the wording related to feedback from the MS. We changed the **Title** of this paper.

**Lines 1-2:** *Response of the Arctic Sea ice-ocean system to melt water perturbations based on a one-dimensional model study*

**Abstract, lines 7-8:** *A one-dimensional coupled sea ice-ocean model is used to investigate how the Arctic Ocean stratification and sea ice respond to changes in melt water.*

**lines 12-13:** *2) Melt water reduced ice melting by 17% by strengthening ocean stratification.*

**Introduction: line 54:** *… the role of melt water in the ice-ocean coupled system of …*

**Section 2.1: line 82:** *… to investigate the influence of melt water in a coupled ice-ocean system …*

**Section 3.2.2 (a): lines 324-325:** *This implies that the presence of melt water inhibits sea ice melting during the melting season.*

**Section 3.2.2 (c): line 394:** *In summary, the above results indicate that melt water always has an inhibitory effect on ice melting during the melting season.*

**Line 397:** *… which is the main reason for the inhibitory effect of melt water on sea ice melting …*

**Conclusions: line 487:** *… that the presence of melt water exerts an inhibitory effect on the process of sea ice melt.*

**2.** Lines 37 to 46 are confusing. I suggest i) the authors clearly define the sources of external and internal freshwater and ii) the authors state their definitions of melt water and river input.

Thanks for the suggestion. We revised some sentences in the **Introduction**.

**Lines 38-44:** *The external freshwater sources of the Arctic Ocean mainly include Pacific inflow, precipitation minus evaporation and river runoff, with a total annual inflow of approximately 9400±490 km³. The annual outflow volume through oceanic gateways, primarily comprising the Fram Strait, Davis Strait, Fury and Hecla Strait, is approximately 8250±550 km³. Thus, the annual net freshwater flux from the external sources into the Arctic Ocean is about 1200±730 km³ (Haine et al., 2015). The internal sources of liquid freshwater mainly originate from the melting and freezing processes. Approximately 13,400 km³ of freshwater freezes during winter, and 11,300 km³ of freshwater enters the ocean through ice melting (Haine et al., 2015).*

**3.** Lines 47 to 57 – Again, I find the wording around feedbacks still very confusing. It is particularly confusing that the authors introduced a new term called melt water feedback (without a proper definition). I think the authors should carefully look at the manuscript to see if the terminology around feedbacks adds to the paper or if it is an unnecessary complication.

We removed some sentences from the fourth paragraph in the **Introduction** and eliminated the wording related to feedback from the MS.

…

**4.** Lines 152 to 157 – How did the addition of this extra freshwater forcing impact the results? Is it possible to quantify this change? Perhaps this can be addressed as a supplemental figure or a sentence in this section.

We quantified this change at stations A1, A6, E2, and E7. In strongly stratified regions, additional freshwater forcing led to a slight shoaling of the ML (Fig. S1a-r), with the impacts on sea ice simulation results being less than 1% (Fig. S2a-c and Fig.S3a-c). However, at the weakly stratified E7 station, the maximum difference in winter MLD reaches several tens of meters (Figures S1v-x), and the additional freshwater forcing in the MWP-0% run resulted in 3% increase in ice melting and 8% decrease in ice formation. (Fig. S2d and Fig. S3d). We added some sentences in **Section 2.3.**

**Lines 157-160:** *We compared the differences between experiments with and without external freshwater forcing at stations A1, A6, E2, and E7. In regions with strong stratification, the presence or absence of external freshwater has little impact on the results. However, in weakly stratified regions, like station E7, the differences are more pronounced (refer to the supplementary file for further details).*

**5.** Lines 184 to 185 – I found this statement confusing. Do you meant that you changed the ML definition based on stratification?

We did not change the ML definition based on stratification. This paragraph explains why we chose $\Delta\sigma$ =0.03 kg m$^{-3}$ as the threshold for calculating the MLD. In order to avoid reader misunderstanding, we removed excessive explanations and directly referenced relevant literature.

We revised a sentence in **Section 2.4**.

**Lines 187-188:** *In this study, the mixed layer depths (MLDs) are calculated as the depth at which the potential density relative to 0 dbar initially surpasses the shallowest sampled density by the threshold criterion of $\Delta\sigma$ =0.03 kg m$^{-3}$, according to previous studies (Toole et al., 2010; Jackson et al., 2012; Peralta-Ferriz and Woodgate 2015).*

**6.** Figure 3a – Please confirm the title of the y axis – is this what was calculated in equation (8)?

The y-axis in Figure 3a is not calculated based on equation (8). The values of the y-axis represent the net value of the ocean surface freshwater flux. We revised some sentences in **Section 2.4** to provide a clearer explanation of Figure 3a.

**Lines 179-183:** *Figure 3a shows the time series of the net ocean freshwater flux, the sum of freshwater fluxes caused by ice melting/freezing and surface freshwater forcing, for the six experiments at station A1, in which the negative value represents freshwater entering the ocean. In this model, the surface freshwater flux caused by ice melting/freezing is on average several tens of times larger than the external freshwater forcing. Therefore, Figure 3a can be regarded as the ocean*

*freshwater flux caused by ice melting/freezing.*

We revised sentences in the **caption of Figure 3** to clarify the meaning of the y-axis.

**Lines 190-191:** *Figure 3: Simulated net ocean freshwater flux and time series of upper 50 m salinity at station A1. (a): Time series of net freshwater flux at sea surface (the sum of freshwater fluxes caused by ice melting/freezing and surface freshwater forcing). ….*

**7.** Figure 7a to f – It is difficult to see what is happening in the upper 20 m. Does it make sense to only show the upper 50 m, as was done for salinity?

Due to the use of inappropriate x-axis scales, Figure 7a-f does not clearly display the temperature in the upper 20m. Since the temperature variations at stations E6 and E7 occur in the upper 120 m, it is incomplete to only show the upper 50 m. Therefore, we modified the x-axis scales for a clearer view of the temperature in the upper 20 m.

[Figure]

*Figure 7: Simulated temperature (top row) and salinity (bottom row) profiles of control runs and MWP runs in mid-August for stations A2, A4, A6, E2, E6 and E7.*

**8.** Lines 350 to 353 – Do the authors have evidence of heat from the NSTM mixing upwards or downwards or is this speculative? Please be clear.

This is a valuable question. We added sentences in the **Section 3.2.2 (a).**

**Lines 357-361:** *As shown in Figure 7a-d, the temperature below 10 m in the MWP-0% run is lower than that in the control run, indicating limited heat transfer to the underlying layers at strongly stratified stations. Conversely, Figure 7f illustrates a well-mixed pattern of water temperature between 0-120m in the MWP-0% run in the station E7. Moreover, the temperature between 60-120m exceeds that of the control run, suggesting a downward mixing of heat that warms the underlying water layers.*

**This is my second review of a manuscript by Zhang et al. titled "A numerical study on melt water feedback in the coupled Arctic Sea ice-ocean system" in which the authors perform 1D numerical systems of the Arctic Sea ice – ocean system to study the negative feedback related to sea ice melt. In particular, they find that the freshwater from melting sea ice reduced ice melt by ~17% and reduced ice growth by up to 40% depending on stratification.**

**I thank the authors for thoroughly answering my previous questions and comments and I think the manuscript is now in a much better shape and close to publishable form. I only have very minor comments and consider this a minor revision.**

Thank you for your comments, which have greatly improved the quality of this MS. In the following, we provide point to point responses (blue text) to the suggestions and revised the MS *(black bold italic)* accordingly.

**Main comment:**

**1.** The fact that the simulations are only 1-year long and are not in equilibrium should be clearly discussed somewhere (maybe in the discussion section). For example, figure 6 shows that in all cases, sea ice is growing beyond the initial conditions. Since for most of the manuscript the authors are looking at anomalies from the control run, this should be okay, but the simulations are still likely sensitive to the initial conditions. As mentioned in the last round, I would have personally preferred control simulations that are in equilibrium, but in any case, it needs to be clearly stated that the simulations are not in equilibrium and therefore you focus on anomalies from the control state.

Thanks for the suggestion. We added some sentences in **Section 3.1.2.**

**Lines 241-243:** *Figure 6a shows that in all control runs, ice is growing beyond the initial conditions, as the model ran for only one year and did not reach an equilibrium state. Nevertheless, it is still reasonable for this study because this paper focuses on the anomalies from the control run by perturbing the melt water.*

**Style & typos**

**2.** Abstract and elsewhere: I think it would be cleanest to use the same number of significant digits. For example, in the abstract I would suggest rounding all the numbers to the full percentages i.e., 17%, 12% and 40%.

Thanks for the suggestion. We rounded all the numbers in the MS to the full percentage.

**Abstract, lines 12-14: 2) … by 17% by strengthening ocean stratification. 3) … by 12% during the winter, while it decreased 43% in …**

**Section 3.2.2 (a), lines 323-324: … by 21.6 cm (~17%), 6.4 cm (~5%), 3.8 cm (~3%), 2.4 cm (~2%) and 1.2 cm (~1%) (averages of all stations) for the ...**

**Section 3.2.2 (b), line 368: … of 21 cm (approximately 12%) was …**

**Conclusions, line 487: … approximately 17% greater than …**

**line 493: … approximately 12% smaller than …**

**3.** Methods: I think the lower boundary condition (and the depth of the lower boundary) are not stated anywhere – please add.

Thanks for the suggestion. we added a sentence in **Section 2.1**.

**lines 84-85: The bottom boundary condition is zero flux, meaning that there is no exchange between the upper water column and the water below 300 m.**

**4.** Figure 11 and 17. state the months used for melting and freezing seasons in the caption.

Sorry for missing this information in the MS. In fact, the duration of the melting and freezing seasons varies among experiments, depending on the dates when ice melting starts/ends in each experiment. We added some sentences in the **caption of Figure 11** to clarify it.

**Lines 331-333:** *Figure 11: … (a) Effective ice thickness change during the melting season. The melting season for each experiment is defined as the period from maximum thickness in May to minimum thickness in September. (b) Effective ice thickness change during the freezing season. The freezing season for each experiment is defined as the period from minimum thickness in September until the end of the simulation.*

**Line 432:** *Figure 17: Same as figure 11 but for stations A3, A6, E2 and E7 of the thinner initial ice experiments.*